# Non-trivial surface states of samarium hexaboride at the (111) surface

Yoshiyuki Ohtsubo [1,2], Yuki Yamashita[2], Kenta Hagiwara[2], Shin-ichiro Ideta[3], Kiyohisa Tanaka [3], Ryu Yukawa [4], Koji Horiba[4], Hiroshi Kumigashira[4,7], Koji Miyamoto[5], Taichi Okuda [5], Wataru Hirano[6], Fumitoshi Iga[6] & Shin-ichi Kimura [1,2]

The peculiar metallic electronic states observed in the Kondo insulator, samarium hexaboride ($SmB_6$), has stimulated considerable attention among those studying non-trivial electronic phenomena. However, experimental studies of these states have led to controversial conclusions mainly due to the difficulty and inhomogeneity of the $SmB_6$ crystal surface. Here, we show the detailed electronic structure of $SmB_6$ with angle-resolved photoelectron spectroscopy measurements of the three-fold (111) surface where only two inequivalent time-reversal-invariant momenta (TRIM) exist. We observe the metallic two-dimensional state was dispersed across the bulk Kondo gap. Its helical in-plane spin polarisation around the surface TRIM indicates that $SmB_6$ is topologically non-trivial, according to the topological classification theory for weakly correlated systems. Based on these results, we propose a simple picture of the controversial topological classification of $SmB_6$.

[1] Graduate School of Frontier Biosciences, Osaka University, Suita 565-0871, Japan. [2] Department of Physics, Graduate School of Science, Osaka University, Toyonaka 560-0043, Japan. [3] Institute for Molecular Science, Okazaki 444-8585, Japan. [4] Photon Factory, Institute of Materials Structure Science, High Energy Accelerator Research Organization (KEK), 1-1 Oho, Tsukuba 305-0801, Japan. [5] Hiroshima Synchrotron Radiation Research Center, Hiroshima University, Higashi-Hiroshima 739-0046, Japan. [6] College of Science, Ibaraki University, Mito 310-8512, Japan. [7] Present address: Institute of Multidisciplinary Research for Advanced Materials (IMRAM), Tohoku University, Sendai 980-8577, Japan. Correspondence and requests for materials should be addressed to Y.O. (email: y_oh@fbs.osaka-u.ac.jp) or to S.-i.K. (email: kimura@fbs.osaka-u.ac.jp)

The coexistence of strong electron correlation and topological order is garnering much attention nowadays because of various peculiar electronic phenomena that are driven by their synergetic effect[1–3]. The strong topological insulator realised in the bulk (3D) Kondo insulator, namely the topological Kondo insulator (TKI)[1], is being extensively considered as a suitable field to study these effects such as non-trivial reconstruction of the topological surface states (TSS) due to electron correlation[4,5] and spin collective excitation, which can break the TSS without time-reversal symmetry breakdown[6].

Samarium hexaboride ($SmB_6$) is a long-known Kondo insulator, which opens the bulk bandgap at low temperature because of the Kondo effect[7]. It is the first material proposed as a candidate for TKI, which hosts metallic TSS coexisting with strong electron correlation[1,8]. To investigate this unconventional TSS, extensive studies that focused on its surface electronic structure were performed[9–14] mainly by using angle-resolved photoelectron spectroscopy (ARPES) and spin-resolved ARPES (SARPES) on the cleaved (001) surface of $SmB_6$. Although the metallic surface states dispersed across the bulk Kondo gap were discovered in TKI, as predicted[9–13], a subsequent high-resolution ARPES study made a counter-claim regarding such TKI assignment by stating that some of the metallic surface states do not disperse continuously across the bulk Kondo gap but accidentally lie at the Fermi level ($E_F$)[14]. Although numerous other studies such as surface-transport[15] and scanning tunnelling microscopy[16,17] strongly suggest the topologically non-trivial nature of $SmB_6$, the detailed surface electronic and spin texture of $SmB_6$ have remained unclear because of this disagreement. Moreover, a peculiar Fermi surface behaviour of $SmB_6$ has been reported recently through the de Haas–van Alphen (dHvA) measurements[18–21]. All groups reported carriers lying at $E_F$ without electrical conduction, but its interpretation, 2D[18,21] or 3D[19,20], is still under debate. Because of these background, it is desirable to elucidate the surface electronic structure of $SmB_6$ and its topological order.

In this work, we report the TSS of a typical candidate for TKI, $SmB_6$, which is observed on the three-fold (111) surface by ARPES. We can determine the topological order on the $SmB_6$(111) surface from the surface Fermi contours (FC) because of the smaller number of inequivalent surface time-reversal invariant momenta (TRIM) and the absence of commensurate and long-range surface reconstructions, as reported for the (001) surfaces[9,14,16]. The metallic two-dimensional state is clearly observed as dispersed across the bulk Kondo gap opening around the Fermi level at low temperature. Its helical in-plane spin polarisation around the $\bar{M}$ point of the surface Brillouin zone (SBZ), which is one of the surface TRIMs, indicates a non-trivial topological order of $SmB_6$. Based on these results, we propose a simple picture of topological-insulating $SmB_6$.

## Results

**A (111) surface of $SmB_6$.** One of the difficulties in determining the detailed surface electronic structure of $SmB_6$ from the ARPES results is its rather complex surface TRIM conformation on the (001) surface. As shown in Fig. 1a, there are three inequivalent surface TRIMs on $SmB_6$(001). While the TSS should appear as an odd number of closed FC enclosing the TRIMs an odd number of times, three such inequivalent TRIMs allow various possibilities regarding the appearance of the TSS[22]. Considering the multiple surface terminations on the cleaved (001) surface[16], it is quite a difficult problem to determine the topological order of $SmB_6$ solely from the electronic structure of the (001) surface. To overcome this problem, the surface electronic structure with a different surface orientation is desired. However, the $SmB_6$ single

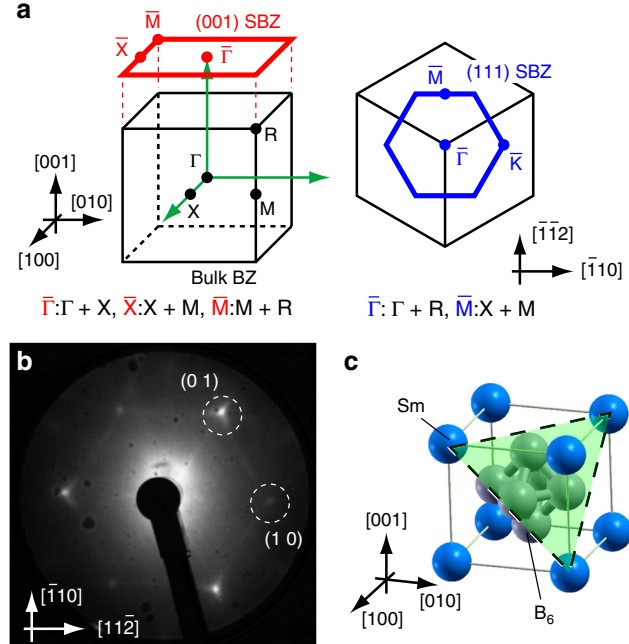

**Fig. 1** Atomic structure of $SmB_6$ and characteristics of the (111) surface. **a** Schematic drawings of the Brillouin zones (BZ). Thin (black) cubes are the 3D bulk BZ with time-reversal invariant momenta (TRIMs), and the thick (red and blue) lines are the first zone boundaries of the 2D surface Brillouin zones (SBZ) with the surface TRIMs. **b** A low-energy electron diffraction (LEED) pattern of $SmB_6$(111) at room temperature. $E_P = 22$ eV. **c** Crystal structure of the $SmB_6$. The dashed triangle indicates the (111) plane

crystal can be cleaved only along (001). Hence, almost no studies have been performed so far on the surface electronic structures with different orientations. Only one set of ARPES data taken from the (110) surface prepared by a similar method to ours has been provided as a preprint[23], but the (110) plane has the same problem as (001); it also contains multiple inequivalent surface TRIMs. The (111) surface of $SmB_6$ is a promising orientation for determining its topological order because there are only two inequivalent TRIMs (right panel of Fig. 1a): one $\bar{\Gamma}$ and three equivalent $\bar{M}$. Note that the other high-symmetry point $\bar{K}$ is not a TRIM. With this simple surface-TRIM conformation, the TSS must appear around one surface TRIM, and thus the determination of the topological order becomes very easy when compared with the previous case. However, no work on the surface electronic structure of $SmB_6$(111) has been reported so far.

In order to obtain the $SmB_6$(111) clean surface, we heated the single crystal up to $1700 \pm 30$ K for 15 min in ultra-high vacuum chambers by using the same method as applied for $YbB_{12}$(001)[24,25]. After heating the sample, one can see sharp and low-background low-energy electron diffraction (LEED) pattern, as shown in Fig. 1b. The three-fold triangular lattice shown by the LEED pattern is consistent with the (111) surface truncated from the simple-cubic lattice (see Fig. 1c). Faint streaks between the integer-order diffraction spots are also seen in the LEED pattern. They would be due to the small area of the facets or long-range surface superstructures without wide commensurate surface areas. It should be noted that the topological order of the material is not influenced by such disordered surface structures and we observed no electronic states related to such surface superstructures in the 1st SBZ, as discussed in the following sections. The (111) surface obtained by this method would be terminated by the Boron clusters, according to the angle-integrated photoelectron spectroscopy[26].

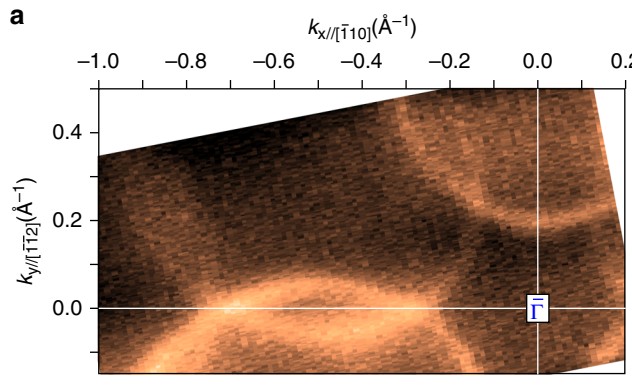

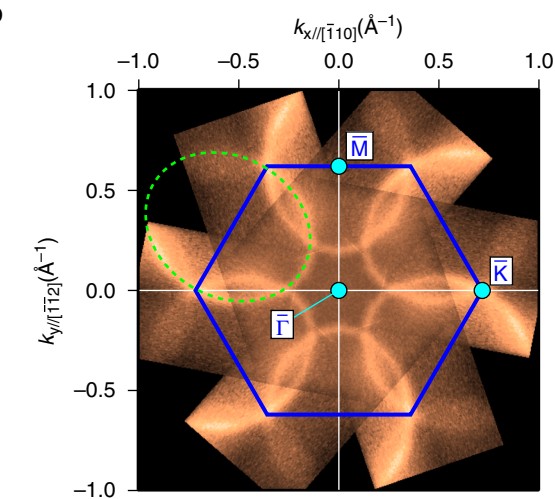

**Fig. 2** Fermi contour obtained by ARPES. The ARPES data were taken with circularly polarised photons ($h\nu = 35$ eV) at 15 K. The ARPES intensities from left- and right-handed polarisations are summed up to show all the states without any influence of circular dichroism. The photon incident plane is slightly shifted from (11$\bar{2}$) because of small misalignment and angle sweep performed for the ARPES scan. This shift is smaller than 15°. **a** Fermi contour with an energy window of 10 meV. **b** Symmetrised Fermi contour based on the three-fold rotation symmetry and time-reversal symmetry. A thick (blue) hexagon is the SBZ boundary of the (1 × 1) surface unit cell

**Surface electronic structure of SmB$_6$(111).** Figure 2a shows the FC around $E_F$ measured with circularly polarised photons at 35 eV. The spectra obtained by using both right- and left-handed polarisations are summed up to avoid circular dichroism. It clearly shows the deformed hexagonal FC enclosing the centre of the SBZ, which is the $\bar{\Gamma}$ point ($k_x = k_y = 0$ Å$^{-1}$). From the symmetrised wide-range overview shown in Fig. 2b, one can find that the deformed hexagon is a part of the ovals enclosing the $\bar{M}$ points, as indicated by the dashed guide. Around $E_F$, no other states are observed by ARPES, indicating that the long-range surface structures observed as faint facets in the LEED pattern (Fig. 1b) play no major role for the surface states around $E_F$. The size of the FCs observed here might be related to the peculiar Fermi surfaces obtained by dHvA measurements[18–21]. For the sake of comparison, we evaluated the sizes of the FCs in Supplementary Note 5.

Figure 3a shows the band dispersions along $\bar{\Gamma} - \bar{M}$ ([11$\bar{2}$]). In order to trace the band dispersion, we took the momentum distribution curves (MDCs) and energy distribution curves (EDCs) as shown in Fig. 3b, c, respectively. The peak positions are plotted with the guides in Fig. 3d so that they could be compared with the 2D data in Fig. 3a. From the MDCs, the highly

dispersive bands, S1 and S2 in Fig. 3d, are clearly observed as the peaks. From the EDCs, less dispersive bands, F at ~0.03 eV and the other, underlying band at ~0.17 eV are observed as the peaks. In addition, the highly dispersive bands S1 and S2 appear in EDCs as broad humps, as indicated by the open triangles in Fig. 3c. Although it is difficult to determine the strict peak positions in the EDCs, the energy region indicated by the bars in Fig. 3c, d have higher intensities than the other EDC spectra (the overlap of them are shown in Supplementary Note 4).

The band lying at the Fermi level, S1, is independent of the incident photon energy range of 15–39 eV, indicating the two-dimensionality from the surface origin. On the left side of Fig. 3d, the projected bulk bands based on the theoretical calculation in ref. [27] is shown as the shaded area. Comparing this with the ARPES data, S1 is out of the projected bulk bands and hence it should be the surface-state band.

For the other band, S2, which is mostly in the projected bulk bands, it is difficult to conclude whether it comes from surface or bulk in the photon-energy range, in which S2 is observable. In this article, we do not conclude the origin of S2, from the surface resonance[28] or bulk Sm-5$d$ bands. The detailed dataset and its analysis is shown in Supplementary Note 2. From the EDCs, it is shown that the F band appears separately from S1. However, it is also difficult to conclude the origin of the F band from the ARPES data. While it appears irrespective to the incident photon energies, the bulk counterpart, the Sm-4$f$ band, is nearly localised and thus it should also show almost no dispersion along the surface normal. The upper edge of the bulk projection in Fig. 3d is slightly lower than F, but the exact values from theoretical calculations, such as the size of the bandgap and the position of the Fermi level, does not always agree with those from experiments. Therefore, the origin of F is not clear from the spin-integrated ARPES. The same analysis was also performed along $\bar{\Gamma} - \bar{K}$ and the similar states to S1, S2 and F were found (see Supplementary Note 3).

The S1 and S2 change their slopes drastically around the crossing point (~±0.2 Å$^{-1}$) with F. These hybridisations are probably driven by the Kondo effect between localised Sm-4$f$ and itinerant Sm-5$d$ states. The upper band S1 clearly disperses across $E_F$ and this band forms the oval FC observed in Fig. 2. The dispersions of the surface state observed here agree well with the expected behaviour of TSS, namely, continuous dispersion across the bulk bandgap and closed FCs around the surface TRIMs. Then, we performed SARPES measurements to examine the spin texture of the FCs, which is regarded as one of the clearest evidence of the topological order of the material.

**Spin texture of the SmB$_6$(111) surface states.** Figure 4a shows the spin-resolved EDCs around the Fermi level measured along $\bar{\Gamma} - \bar{M}$ at 20 K. The spin polarisation along [$\bar{1}10$] and the in-plane orientation perpendicular to $k_{y//[11\bar{2}]}$ were resolved. From the EDC Cuts 1 to 4, one can easily find that the spin-polarised feature towards [1$\bar{1}0$], indicated by the negative spin polarisation values, disperses from +0.03 to −0.03 eV across $E_F$, which is consistent with the metallic dispersion of the surface band S1. At the opposite side of the SBZ (Cut 5 in Fig. 4a), the opposite spin polarisation towards [$\bar{1}10$] (positive spin polarisation) is also observed. Such spin inversion according to the sign inversion of $k_y$ indicates that these spin polarisations conserve the time-reversal inversion symmetry. The spin polarisation value at positive $k_y$ is nearly twice that at negative $k_y$. This would be due to the lack of mirror plane normal to $\bar{\Gamma} - \bar{M}$, as shown in the atomic structure and the LEED pattern in Fig. 1b, c, respectively.

One may doubt that if the energy resolution of the current SARPES setup, ~30 meV, is enough to trace the spin polarisation

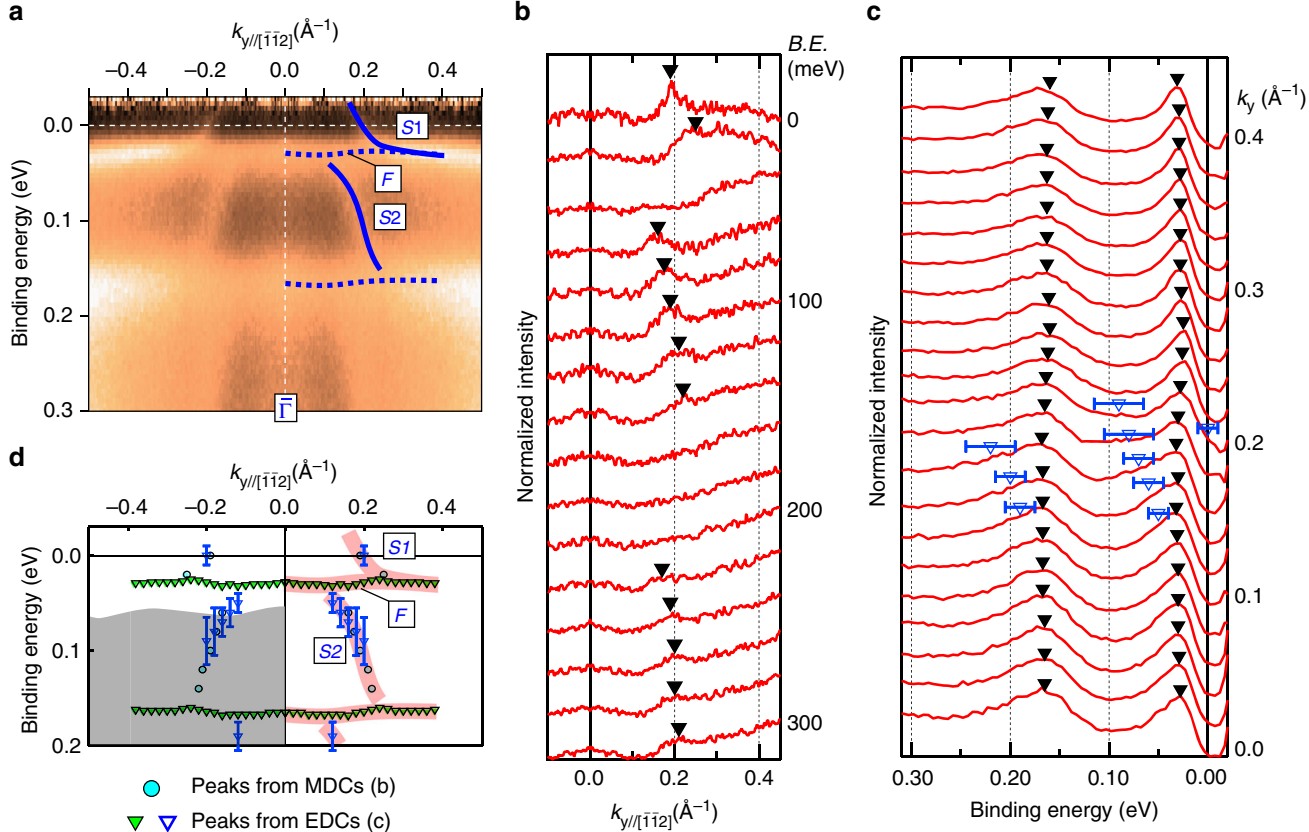

**Fig. 3** Band dispersions of SmB$_6$(111) around the Fermi level. ARPES data were taken with the same condition as Fig. 2. **a** ARPES intensity plots along $\bar{\Gamma} - \bar{M}$ symmetrised with respect to $\bar{\Gamma}$ ($k = 0$ Å$^{-1}$). ARPES intensities are divided by the Fermi distribution function convolved with the instrumental resolution. **b**, **c** ARPES **b** momentum distribution curves (MDCs) and **c** energy distribution curves (EDCs) taken from the 2D data shown in (**a**). Triangle markers indicate the peak positions. The open triangles with error bars in (**c**) are the energy positions of broad features. The width of the bars is explained in the text. **d** 2D plot of the peak positions in (**b**, **c**). The bars with open triangles are the same as those in (**b**). The shaded area in the left side is the projected bulk bands from ref. [27]. Fat curves are the traces of the peak positions. These curves are copied on (**a**)

of the surface states or not, since this resolution is close to the total size of the energy window where $S1$ is visible. However, the SARPES data showed the clear spin polarisations well above the noise level evaluated by the standard statistical errors and evident spin-polarised peaks consistent with the dispersion of $S1$ obtained from the spin-integrated data (Figs. 2 and 3). They proved that the current SARPES data is enough to trace the spin polarisation of the surface states, without any ambiguity.

The spin-resolved EDCs indicate that the spin polarisations of the deeper features, from 0.15 eV at Cut 1 ($k_y = +0.29$ Å$^{-1}$) to 0.04 eV at Cut 4 ($k_y = +0.17$ Å$^{-1}$), are opposite to $S1$. These deeper features correspond to $F$ and $S2$ observed by the spin-integrated ARPES. The similar feature was also observed in the opposite side of the SBZ (Cut 5: $k_y = -0.25$ Å$^{-1}$). These polarisation values are also above the estimated errors as shown by the error bars in Fig. 4a. If one assumes the $S2$ and $F$ to be the surface bands, such spin polarisations could be understood as a result of space inversion asymmetry in the surface layers. On the other hand, even if $S2$ and $F$ come from bulk bands, such spin polarisation can appear because of the spin-dependent reflectivity of the Bloch waves at the surface[29]. Therefore, we can neither determine the origin of $S2$ and $F$, from surface or bulk, by the spin resolution. From the dispersions of $S1$ and $F$, they apparently degenerate with each other close to $\bar{M}$. This behaviour might be the Kramers degeneracy between $S1$ and $F$ as the surface bands, or the surface band $S1$ merging into bulk bands $F$. Anyway, to verify these assumptions, dispersions of them in the vicinity of $\bar{M}$

with spin resolution should be measured. Such measurement was not possible in this work because of the limited energy resolution of SARPES; even far away from $\bar{M}$ (around 0.4 Å$^{-1}$), it is impossible to resolve $S1$ and $F$. To examine this assumption, the higher energy resolution in SARPES is desirable.

The spin polarisations of the surface states along $\bar{\Gamma} - \bar{K}$ were measured by the spin-resolved MDCs as shown in Fig. 4b. The MDC peak heights along $\bar{\Gamma} - \bar{K}$ are nearly equal to each other, reflecting the presence of the mirror plane normal to $\bar{\Gamma} - \bar{K}$. Although the MDC peak shape is not symmetric along $\bar{\Gamma} - \bar{K}$, one can find its peaks at $k_x \sim \pm 0.3$ Å$^{-1}$, which is consistent with those of metallic oval FCs shown in Fig. 2. The origin of asymmetric peak shape and its influence to spin polarisation is shown in Supplementary Note 8. The in-plane spin polarisations along $\bar{\Gamma} - \bar{K}$ are consistent with those shown in Fig. 4a. In addition, clear out-of-plane spin polarisations are also observed along $\bar{\Gamma} - \bar{K}$, as shown in the lower part of Fig. 4b, whereas such spin polarisations are almost negligible along $\bar{\Gamma} - \bar{M}$ (see Supplementary Note 6). Along $\bar{\Gamma} - \bar{K}$, the in-plane and out-of-plane spin polarisations of the photoelectrons are nearly equal to each other. Such out-of-plane spin polarisations would be due to the coupling between spin–orbit interaction and the valley degree of freedom around $\bar{K}$, where three-fold rotation symmetry appears without time-reversal symmetry, as observed in Tl/Si(111)[30] and transition-metal dichalcogenides[31].

Based on the SARPES spectra, we depicted a schematic drawing of the spin texture of the metallic surface states on

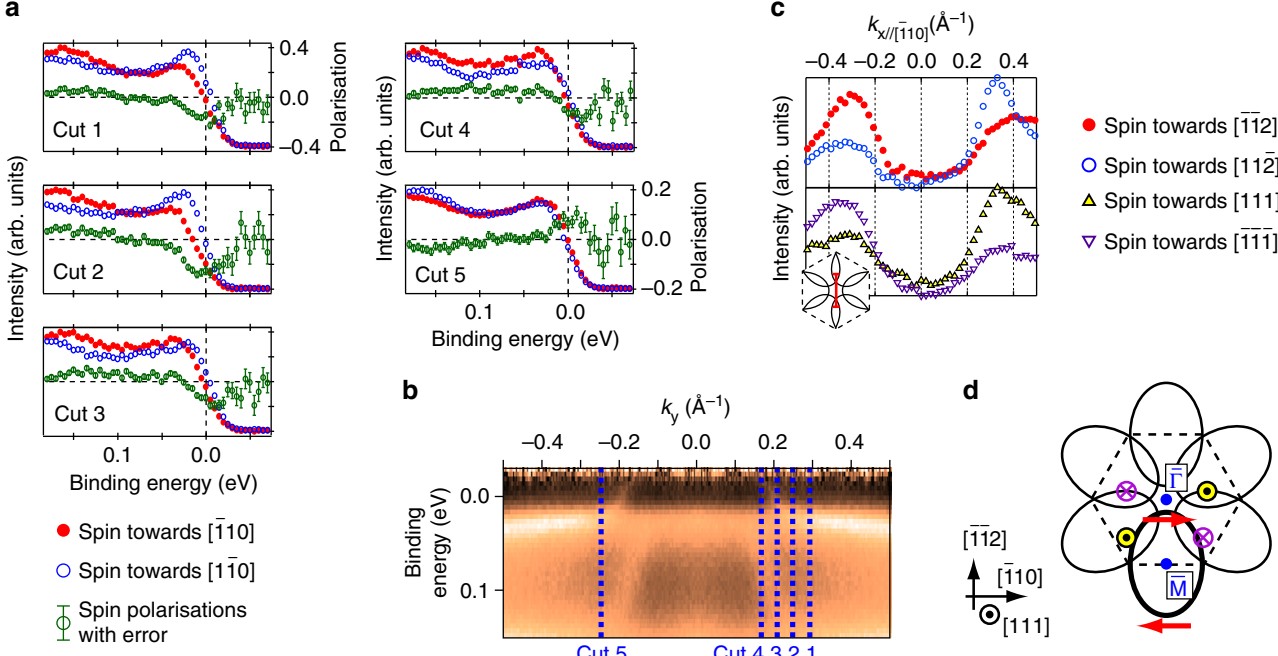

**Fig. 4** SARPES spectra. **a, b** SARPES spectra taken with linearly polarised photons ($h\nu = 26$ eV) at 20 K. Detailed experimental geometries are shown in Supplementary Note 1. **a** Spin-resolved energy distribution curves (EDCs) around $k_F$ together with spin polarisations. Errors of spin polarisation values are standard statistical errors from photoelectron counting. **b** Spin-resolved momentum distribution curves (MDCs) along $\bar{\Gamma} - \bar{K}$ at $E_F$. Inset is the schematic drawing of the Fermi contour together with the $k$ range where the spin-resolved MDCs were observed. **c** The same as Fig. 2c to indicate the positions where the spin-resolved EDCs in (**a**) were observed. **d** A schematic drawing of the spin texture of the Fermi contours formed by topological surface states on SmB$_6$(111). The arrows and circles with crosses and dots inside depict the in-plane and out-of-plane spin polarisations, respectively

SmB$_6$(111) in Fig. 4d. As shown in the figure, the oval FC enclosing $\bar{M}$ has clockwise spin polarisations along the in-plane orientations and finite out-of-plane ones away from the surface mirror plane parallel to $\bar{\Gamma} - \bar{M}$. Such non-zero component along the out-of-plane orientations is natural for topological states on the surfaces with three-fold rotational symmetry, e.g. those on Bi$_2$Te$_3$[32]. The detailed discussion about the role of surface symmetry operations to the spin polarisations is shown in Supplementary Note 7.

The whole spin texture, both in-plane and out-of-plane ones, qualitatively agrees well with a recent theoretical calculation[27], supposing the negative winding number $w = -1$. One should be careful for such comparison between theory and SARPES data, because sometimes spin polarisation of photoelectrons occurs artificially due to photoexcitation process and/or spin–orbital entanglement[33,34]. However, it should be noted that the spin polarisation whose sign inverts with respect to time inversion, as shown in Fig. 4, cannot appear from spin-degenerate states even if such artificial spin polarisations occurred. Although it is sometimes shortcoming to connect the observed spin polarisation of the photoelectrons to those of the initial states directly, it is evident that the initial states, $S1$, $S2$ and $F$, are somehow spin-polarised and its sign inverts according to the surface symmetry. This information is enough to discuss the topological order of the sample from its surface states, as shown in the following.

## Discussion

Based on the spin-polarisation and shape of the FCs, the topological order of SmB$_6$ is calculated. In order to obtain the topological order of a material from its surface states, one has to obey the following procedure[22]:

(i)     Count up the FCs enclosing surface TRIMs.

(ii)    Observe them by SARPES to check if they are spin polarised or not. The number counted in (i) is doubled for the spin degenerate states.

(iii)   Examine the summed number. If it is odd, then the sample is a (strong) topological insulator. If even, the sample is normal, trivial insulator.

On SmB$_6$(111), the FCs enclosing $\bar{M}$ appears three times (i); note that there are only three (not six) inequivalent $\bar{M}$ points because of the translational symmetry by the surface reciprocal lattice vectors. Since all the FCs here are spin polarised (ii), the total count in this case is three, the odd number. According to this calculation, SmB$_6$ is determined to be a topological insulator, without any ambiguity. Note that the same procedure is difficult to be applied to the (001) surface of SmB$_6$, since the number of FCs are still under discussion[10–14]. At first glance, this conclusion appears to conflict against a recent high-resolution ARPES data on (001)[14]. However, they could be reconciled by an interpretation of the FCs observed on (001). Detailed discussion on this point is shown in Supplementary Note 9. The other important point is that the detailed origin of the surface states play no major role for the classification discussed above. It is because the topological classification is merely from the number of the spin-polarised FCs[22,35]. In other words, if the odd-number of FCs were made by surface states derived from many-body resonance for example, the insulating substrate would be nothing but a topological insulator. We discuss the possible origins of the surface states observed in this work in Supplementary Note 10.

The dispersion of TSS ($S1$) in this work does not show any Dirac point. Instead, it shows quite a light velocity of ~0.8 eV Å (see Supplementary Note 4 for its estimation) only around the Fermi wavevector ($k_F$). Away from $k_F$, $S1$ becomes quite heavy with almost no dispersion at the binding energy of ~30 meV. Such behaviour agrees well with the theoretically expected TSS

dispersion modified by the Kondo breakdown[4]. The expected Fermi velocity in ref. [4] is ~0.3 eV Å, showing an agreement of the order of magnitude with the experimental value above. Further theoretical work taking the large size of FC overlapping with each other and/or out-of-plane spin components into account might be applicable to solve this factor 2–3 discrepancy.

At last, we would like to state the limitation of the current work. The topological classification procedure[22] assumes weakly correlated insulator. Therefore, we cannot exclude a possible violation of such simple topological classification by strong electron correlation. To the best of our knowledge, such work has never been published so far. However, once such discovery has been achieved, the topological classification performed in this work should be revisited. The other limitation is the bulk electronic structure of SmB$_6$ at 15–20 K, where we made the measurements. It is commonly regarded that the rather wide activation gap (~20 meV) opens and the bulk electronic structure transforms to an insulator in this temperature range[36]. However, a recent ARPES study[14] has claimed that the bulk gap is still closed there, attributing the other small gap (3–5 meV, which opens below 10 K) is the real gap. Although no experimental data supporting the claim above, the bulk band surviving at $E_F$ around 20 K, has been reported so far, we have to admit that the topological classification in this work becomes nonsense in this temperature range, if this claim is correct. Note that the claim in ref. [14] is not the remnant bulk carriers thermally excited across the gap (~20 meV) but the firm band without any gap across the Fermi level. It should also be noted that the peculiar Fermi surface observed by quantum oscillation experiments[18–21] are not relevant to this possibility, because they were performed in much lower temperature range.

In summary, the TSS of a TKI candidate, SmB$_6$, was clarified with regard to the different surface orientation from the earlier works, the three-fold (111) surface, by ARPES, in this work. The metallic two-dimensional state was clearly observed as dispersed across the bulk Kondo gap opening at the Fermi level at low temperature. Its helical in-plane spin polarisation around the $\bar{M}$ point of the SBZ, which is one of the surface TRIM, provided the evidence of the non-trivial topological order of SmB$_6$. Based on these results, we propose a simple picture of topological-insulating SmB$_6$ to be a fascinating groundwork to study peculiar electronic phenomena such as the synergetic effects with strong electron correlation.

## Methods

**Sample preparation.** Single crystalline SmB$_6$ were grown by the floating-zone method by using an image furnace with four xenon lamps[37,38]. The sample cut along the (111) plane was mechanically polished in air until a mirror-like shiny surface was obtained with only a few scratches when observed under an optical microscope (multiple 10× magnification).

**ARPES and SARPES experimental setup.** The ARPES measurements were performed with synchrotron radiation at BL7U SAMRAI[39] of UVSOR-III and BL-2A MUSASHI of the Photon Factory. The photon energies used in these measurements ranged from 18 to 80 eV. SARPES measurements were performed at HiSOR BL9B ESPRESSO[40] with linearly polarised photons at 26 eV so that the photoelectron spin polarisation due to the circularly polarised photons are excluded[41]. A pair of the very low energy electron diffraction (VLEED) detectors enable the three-dimensional detection of the spin polarisations[42]. The effective Sherman function of the spin detector was set to 0.3 and the acceptance angle for the detector was ±1.5°. The energy resolutions of the spin-integrated and SARPES in this work were ~15 and ~30 meV, respectively. The energy resolutions and photoelectron kinetic energies at the Fermi level $E_F$ were calibrated using the Fermi edge of the photoelectron spectra from Ta foils attached to the samples. The detailed experimental geometries are shown in Supplementary Note 1.

The SARPES spectra were measured four times as $I_p^1$, $I_n^1$, $I_n^2$, $I_p^2$ with this order, where $I_p^i$ and $I_n^i$ ($i = 1, 2$) are the raw spectra obtained from the VLEED detector with positive and negative target magnetisation, respectively. Then, we got $I_p = I_p^1 + I_p^2$ and $I_n = I_n^1 + I_n^2$. By this procedure, we compensate the time-dependent degradation

of the surface states as well as the decay of incident photon flux (proportional to the beam current of the storage ring of HiSOR). The spin polarisation of the SARPES spectra is calculated by $P = (I_p - I_n)/S_{eff}(I_p + I_n)$, where $P$ is the spin polarisation shown in Fig. 4 and $S_{eff}$ the effective Sherman function. $S_{eff}$ is calibrated by the spin polarisation of the well-known surface state[40]. The errors of $P$ is calculated as the standard statistical error. Then, the spin-resolved spectrum $I_\pm$, which are shown in Fig. 4, is calculated by $I_\pm = (I_p + I_n)(1 \pm P)/2$. For the SAPRES spectra in Fig. 4, no normalisation nor smoothing procedures have been applied.

## Data availability

The datasets generated during and/or analysed during the current study are available from the corresponding author on reasonable request.

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

## Acknowledgements

The authors thank T. Nakamura, Y. Takeno and C. Wang for their support during general experiments. The authors acknowledge S. Wu for his support during the experiments on BL9B at HiSOR. Part of the ARPES measurements were performed under UVSOR Proposals 29-553 and 30-577, Photon Factory Proposals Nos. 2015G540 and 2017G537, and HiSOR Proposal No. 17AG017. This work was also supported by the JSPS KAKENHI (Grant Nos. JP15H03676, JP17K18757 and JP23244066).

## Author contributions

Y.O., Y.Y. and K.H. conducted the ARPES experiments with assistance from S.-i.I., K.T., R.Y., K.H. and H.K. as well as the spin-resolved APRES assisted by K.M. and T.O. W.H. and F.I. grew the single-crystal samples. Y.O. and S.-i.K. wrote the text and were responsible for the overall direction of the research project. All authors contributed to scientific planning and discussions.

## Additional information

**Competing interests:** The authors declare no competing interests.

