## [Peer Review File · Nature Communications]

Reviewers' comments:

Reviewer #1 (Remarks to the Author):

The manuscript by Ohtsubo et al. presents an ARPES study of the (111) surface of SmB6. SmB6 is predicted to be a topological Kondo insulator, however the determination of the topologically non-trivial nature of this material is still elusive and debated. Common problems are the unreliable sample quality, the small size of the Kondo hybridization gap, the polarity of the cleaved surface and the surface reconstruction. The authors address the issue by measuring the (111) surface instead of the most studied (100). The authors claim that the Fermi surface of the (111) plane present two dimensional features compatible with topological surface states, they further support the conclusion by performing spin-polarized ARPES and revealing an apparent spin texture of the surface states. In this manuscript Ohtsubo et al. claim this study conclusively proves the existence of the non-trivial topological insulator phase in SmB6.

While this approach is novel and timely and could provide important insights in the long-standing issue of topological order in Kondo systems, the manuscript in its present form does not provide a convincing argument for the conclusions drawn, and therefore I do not recommend its publication in Nature communication at this stage or in this form.

I might support publication only after extensive and convincing revision of the paper, addressing all the points I am listing below.

In the following are listed the main points / questions that I believe should be addressed:

1 – Properties of the (111) surface.

As stated in the manuscript the commonly studied (100) surface poses several experimental challenges: Namely the surface reconstruction, the atomic termination and domains and surface polarity [Kim et al., PRB 90, 075131, Zhu et al., PRL 111,216402, Denlinger et al. Arxiv. 1312.6636]. Can the authors comment and describe the expected and obtained structural and electronic properties of the (111) surface in respect to these points? What is the surface termination and is the surface polar? Previous works have demonstrated how surface polarity gives rise to trivial surface states on the (100) surface [Zhu et al., PRL 111,216402] and how seemingly in-gap states arise from surface resonance on different terminations [Hlawenka et al. , Nat. Comm (2015)]. The characteristics and advantages of the (111) surface with respect to these arguments should be discussed and demonstrated in the paper (e.g. DFT, see below).

2 – The labelling of the bands in Fig. 2 and in the text is unclear.

Bands are labelled as S1, S2 and S1' but a clear assignment of these bands is missing and only a guide to the eye is shown in the Fig.2. Can the authors better define the bands they see with photoemission? The bulk band structure of SmB6 is well-known, how do the labelled features relate to it?

The text is very obscure in this regard, I can infer that S2 is the bulk d-f band derived from the hybridization of the Sm 4f and 5d states. S1 is the putative surface topological state as it's dispersing across the gap. The definition of S1' is even more vague and it is later assigned to a Kramer pair of S1, but it rather looks like intensity from the f-band. I encourage the authors to clarify the description of the observed band structure in terms of expected bulk and surface bands.

3 – The text does not clearly state the bulk or surface nature of the observed band structure.

Fig.S2 shows two Fermi surfaces obtained at different photon energy in order to demonstrate the surface nature of the states at the Fermi level. Can the authors provide a more quantitative analysis of the k_z dispersion? On first sight it looks to me like the area of the Fermi surface pockets is different from fig.S2a, S2b and Fig.2b by comparing the size of overlap between neighboring ovals. This is probably just a resolution effect but a more quantitative analysis is needed for a convincing argument. I encourage the authors to show the k_x dispersion as a function of photon energy in the whole 18-35 eV range measured, and to show a comparison

between the surface states within the Kondo gap and the bulk states below it.

Finally I think the paper would greatly benefit from a comparison with DFT calculated band structure that would help in the assignment of the bands and with the surface/bulk distinction increasing the value of the paper.

4 – I find the spin-ARPES curves shown in Fig.3 very confusing.

In particular:

- 1 – The spin EDCs in Fig.3c do show asymmetry for the in-gap state. However there seems to be a significant spin polarization on the valence bands as well (below -0.03 eV). Overall there seem to be a significant asymmetry in all states shown in the EDCs, down to below 0.15eV. The authors should comment on the analysis procedure for the spin ARPES data, in particular how the curves are renormalized, how the polarization is calculated, and whether the curves shown are raw data or renormalized by the Sherman function (spin-data); error bars should be shown. The spin polarization switches at deeper binding energy. In the text this is attributed to the S1' state as "Kramer partner" of S1. This is non-conform with the definition of "Kramer" partner that is found at the time reversal point in the Brillouin zone. If the authors mean the state of opposite spin found at the same k-point (i.e. upper vs lower branches of the topological state) it should be justified how this state can exist seemingly within the valence band. Also, the S2 state at higher binding also shows spin-polarization, however this should be a bulk state (if I understood correctly) and should not be polarized.
- 2- From fig.3e, the in-plane spin polarization appears to be chiral and tangential around the pockets. However, in SM figS3 it appears that the spin polarization along Gamma-M is along the [-1 -1 2] direction, while it should be towards [-110] according to the main text. Is this a mistake in the legend?
- 3- The out of plane polarization shows indeed a strong asymmetry along Gamma-K in Fig.3b, but its spin texture is not clear from the text and Fig.3e. Although an out of plane polarization is not symmetry forbidden along Gamma-K, in the simplest terms I would expect that the out of plane polarization cancels out along this direction because the two oval pockets are degenerate and should carry opposite spin polarization. How is the out of plane signal justified? Is it an effect of photoemission cross section? Can the authors elaborate on this and show it more clearly in the Fig.3e?

Overall the spin-ARPES analysis seem shortcoming and does need a more thorough discussion. I recommend the authors to review the spin-ARPES analysis commenting also on possible spurious effects of the technique, as shown in previous works [Heinzmann et al. J. phys. Cond. Matt. 24, 173001, Jozwiak et al., Phys. Rev. B 84, 165113, S. McKeown Walker et al., Phys. Rev. B 93, 245143]

5 – I am a bit puzzled by the discussion paragraph: The authors refer to the paper { Hlawenka et al, Nat. Comm (2015) , ref.13 in manuscript } and write that the "slope of the "umklapp" band is different from that of its counterpart". This does not seem obvious to me from the referenced paper and the authors should better/quantitatively justify this claim. Finally throughout the paper the authors state very strongly that their findings are a conclusive prove of the topological nature of SmB6, I think even if all my previous points are appropriately addressed this claim is too strong as the authors themselves discuss in the last paragraph stating that "the current ARPES result shows agreement and disagreement with the two conflicting report" and "comparison between the ARPES data from multiple surface orientation" ... "and other bulk sensitive methods would be helpful in solving this complicated problem"

6 – Overall, I find the paper lacking in overall presentation and both readability and figures should be improved for publication in Nat. Comm.

Reviewer #2 (Remarks to the Author):

I have read with great interest the manuscript by Ohtsubo and coworkers. The debate on SmB6 is on two levels. The first level is whether SmB6 is a topological insulator not. There is a lot of evidence (theory, transport etc.) that indeed SmB6 is a topological Kondo insulator. The second level of the debate is purely related to ARPES data and the question is whether the near-EF electronic states around $X_{\bar{}}$ –seen by all ARPES studies- are the long-sought topological states or not. Here the answer is not obvious, as those states have been attributed to surface analogues of the bulk conduction band [Hlawenka ncomms 2018]. The attribution of those states to (surface analogues of) the bulk d bands would mean that the Kondo gap is very small ($<5\text{meV}$) which would give a better agreement with transport in terms of its size but it would lie some 30meV below EF, presumably due to band bending. On the other hand, if these states were the long-sought topological states the Kondo gap would be larger than 30meV and it would lie at the Fermi level.

The work of Ohtsubo et al. contributes to that second level of debate. The preparation and data acquisition on a clean SmB6 (111) surface is very interesting and it has not been reported before. I congratulate the authors for these data. In that sense, the results of this work should be of course accessible to the scientific community but for the reasons that I will shortly explain I am against publication with the present conclusions.

The so-far debatable ARPES data consisted of electronic states at $X_{\bar{}}$ that were reminiscent of the bulk Fermi surface contours (bulk contours = ellipsoids at bulk X points). These $X_{\bar{}}$ states were however shown to be spin-polarized [Xu ncomms 2014] favoring the interpretation of topological states. In this sense, the new data by Ohtsubo et al. does not give new information even if it was acquired on a new surface. To make my point clear: the bulk ellipsoids at X when projected on the (111) surface Brillouin zone would exactly give the contours seen by Ohtsubo and coworkers. The situation is reminiscent of the 2DEG observed in bare surfaces of transition metal oxides. For example, the 2DEG observed on different surfaces of SrTiO3 -that is in fact a confined version of the bulk conduction band- is projected on the (001) surface [Santander Nature 2011], the (110) surface [Rodel PR applied 2014] and the (111) surface [Walker PRL 2014] giving each time different contours which are in perfect agreement with the corresponding projection of the very same bulk state. Why don't the authors consider this possibility for SmB6? A counterargument would be the observed spin polarization but the existence of spin polarization already known since 2014 for the very same states on the (001) surface of SmB6.

Concerning the SARPES data: The observation of spin polarization is very important but SARPES data has an energy resolution (30 meV) that is equal to the total range of the energy dispersion of the S1 state. I am aware that this value is very competitive in terms of resolution for SARPES but in my opinion the small energy range of dispersion for the near-EF states in SmB6 makes SARPES unsuitable for this specific system. I also have some specific comments on these data:

- In Fig. 2d, cuts 1 and 2 do not seem to go through the S1 state. Yet, the spin polarization is clear. How can the authors explain that? Could it be that the S1prime is polarized INSTEAD of S1?
- The authors state that S1prime and S2 have opposite polarization to S1. How can this be reconciled with what we know about SmB6? S1prime is a flat band of f-origin while S2 is the highly dispersing d-band. These features are well known and spin polarization is not expected for neither of the two.

To conclude: Although the data is interesting and its acquisition must have been very challenging I cannot recommend publication because the authors are guiding the reader towards the conclusion of topological states with no new insight with respect to what we so far knew (i.e. the FS contours are new but well expected and the spin polarization had been already observed). I am not suggesting a bare reformulation of 2-3 sentences in the manuscript but a fundamental rewriting of

the discussion –and not only– to include all different possibilities (confined bulk states vs. topological states) and to mention the limitations of SARPES data. I believe that the reader should have the right to choose which interpretation is the best without being influenced. In contrast to the authors' statement, in my view, the "second level" of the debate on SmB6 is far from being concluded.

Reviewer #3 (Remarks to the Author):

This manuscript reports on Angle-resolved photoemission and spin-ARPES of in-gap states for the prepared (111) surface orientation of SmB6. Complementary to previous measurements on the (001) cleaved surface and also prepared (110) surface, the authors claim the (111) surface has advantages for establishing the topological index.

The (111) surface ARPES of SmB6 is novel and the improved spin-ARPES technique resolution from 30 meV to 15 meV for this system is a welcome technical advance. Hence the results here are highly publishable in Nature Communications. However there are numerous corrections, clarifications and improvement recommendations to be discussed:

In-gap Fermi surface size

The most striking and unexpected experimental result concerns the large size of the in-gap state elliptical contours in comparison to the in-gap states for the (100) and (110) surfaces. For instance Ref. 21 provides some size numbers in terms of dHvA orbit frequencies (discussed more later): For the (100) surface the in-gap state orbit size is $F=3.5$ kT and for the (110) surface the smaller in-gap contour is 2.0 kT. In contrast, here for the (111) surface $k_a=0.35$ and $k_b=1.2*0.35=0.42$, the dHvA frequency is $F=10.47*Area = 10.47*\pi*0.35*0.42 = 4.84$ kT. Also for comparison Ref. 21 gives the Sm 2.5+ mixed valent and LaB6 trivalent bulk 3D ellipsoid extremal orbit sizes of 4.5-6.0 kT and 7.5-10.0 kT, respectively.

To better understand this large in-gap state contour result, it would be instructive if the authors provided supplementary information comparing the in-gap states to the bulk d-states. I.e. (a) present and quantify a constant energy map from the deeper binding energy bulk d-band just below the highest f-state (approx. -50 meV), and (b) quantify the k_F -shift between the in-gap states and the extrapolation of the d-band states to EF.

For (a) does it have an even larger overlap of elliptical contours? Also do the d-states outside the gap exhibit any 3D variation in size between 19, 26 and 35 eV? For (b) the (100) surface numbers provided by Ref. 21 indicate a k_F -shift as large as 0.11-0.13 (inverse angstroms). Here visually from Fig. 2(c) the extrapolation of the d-band dispersion appears to have a smaller k_F -shift of about 0.08.

Providing such quantitative numbers allows the authors or readers to then contemplate some deeper meaning to the difference with the (100) surface. Is it just a directional projection effect, or does it reflect a difference in a Kondo surface breakdown scenario as proposed by Alexandrov PRL 114, 177202 (2015) and Peters, Ref. 4 or something else? E.g. comparison of a 2D state to the projection of bulk 3D ellipsoid to the surface can also be tricky. For instance Koitzsch et al. Nat. Comm. 7, 10876 (2016) measure bulk-sensitive (111) FS contours of trivalent CeB6(111) and find the contours to just touch each other consistent with the 3D ellipsoids and NOT overlap as might expect projected to the surface. In this illusory sense, the 2D in-gap state of the 2.5+ valence system here is even larger than the trivalent CeB6 FS contour.

Overlap of topological states

Is the overlapping of topological surface states from separate neighboring TRIM points a new unprecedented result not reported earlier in the literature? The authors should comment on this and discuss. Certainly there are reported complex interaction effects of the dual Dirac cones when two bulk TRIM points project onto the same surface BZ point, but that is a different scenario than the result here.

Does one expect any interaction between the overlapping surface states? If so, one would imagine that surface state electron orbits might not be broken down into smaller sized orbits of the overlap region and the central warped hexagon. What dHvA frequency corresponds to these two smaller orbits? Or can it be argued that there should be no interaction, and that since the in-gap states are single-spin states are they allowed as Fermions to share the same momentum point if their spin states are orthogonal?

Spin-ARPES

(1) A crucial citation that is missing is Baruselli and Vojta, PRB 93, 195117 (2016) which provides a theoretical prediction for the spin texture of the topological states for the (100), (110) and (111) surfaces of SmB6. In particular they predict opposite clockwise/counterclockwise in-plane helicities of the spin for different winding numbers ($w=\pm 1$) for all the surfaces as well as a specific variation of the (111) out-of-plane component of the spin for the case of $w=+1$. The authors here also have in-plane and out-of-plane results and should compare and discuss their measured spin texture to that prediction.

The schematic of the (111) spin-ARPES results in Fig. shows a clockwise spin-helicity that would correspond to the Baruselli parameter $w=-1$. This appears to agree with the claim that the (001) spin-ARPES result also agrees with $w=-1$.

(2) The authors use two different geometries to measure the in-plane spin-polarization at the ellipse tips. In Fig. 3(a,c,d) the tilt angle is used to access $k_y = \pm k_F$ and the spin-polarization is observed towards the (-110)/(1-10) directions which is tangential to the ellipse tip. In the supplemental Fig. S3(a), the sample azimuth is rotated and the polar angle is used to access $k_x = \pm k_F$ at the ellipse tips and strong spin-polarization is observed towards the (-1-12) or (11-2) horizontal directions. However, due to the azimuth rotation, this direction is now "radial" to the ellipse. How am I supposed to understand both tangential AND radial in-plane spin-polarization component results between Figures 3(a,c,d) and S3(a)?

(3) Another obvious citation that is missing is another spin-resolved ARPES measurement of SmB6 by Suga et al. JPSJ 83, 014705 (2014). In that study, they failed to reproduce the spin-polarization of the (001) surface in-gap states reported by Xu et al. [Ref 12], but along the way they did report strong spin-polarization of the SmB6 f-states using 35 eV circular polarized excitation (and none for linear polarization). I see that the spin-ARPES measurements here are for linearly polarized 26 eV photons. I believe it is useful for the authors to cite this other work and comment on how or how much they are able to avoid Sm 4f-state spin-polarization contributions?

(4) While the authors characterize spin-ARPES results as "regarded as indisputable evidence of the topological order", in fact there exists an example of spin-ARPES claims of topological order in surface states of another hexaboride, YbB6 (Xu, arXiv:1405.0165), that was in fact disputed (Kang, PRL 116, 116401 (2016)). I do see words in this paper about being able to eliminate geometric contributions to the spin-polarization. Do the authors have a particular citation that describes the potential instrumental/geometrical artifacts in the spin-ARPES technique?

Relation to dHvA controversy

The abstract and introduction promise new insight to the dHvA controversy. It was then disappointing to find that this boiled down to a very incomplete (and also incorrect) comparison of the size of the large ARPES in-gap state elliptical Fermi contour to the the large alpha-band trivalent LaB6-like dHvA orbits reported by the Cambridge group (Ref. 18,19). The authors note that (a) their (111) 2D surface state orbit size is much larger than the small orbit size by the Michigan group (claimed to be 2D and originating from (110) surfaces), and (b) that the 2D ARPES contours have semi-agreement to the large Cambridge orbits (which are claimed to be 3D). There is much to be discussed about problems with the presentation of their comparison to dHvA:

(1) The introduction statement "ref. [17] reported small 2D shapes whereas ref [18, 19] reported large 3D shapes" is incorrectly stated. Both dHvA groups agree experimentally in the presence of a strong amplitude small-sized orbits (with $F_{\min}=290T$), but differ in their interpretation of 2D origin (Michigan) versus 3D origin (Cambridge, $F_{\max}=700T$). Ref. [18,19] "additionally" report multiple higher frequency weak-amplitude orbits greater than 1 kT including trivalent-like 10 kT sized 3D orbits.

Hence the ARPES large ellipse orbit size disagrees with (or has no correspondence) to the main quantum oscillation result of "both" dHvA groups. As stated earlier, the authors could provide a quantitative comparison of a potential smaller ARPES overlap orbit to the strong small-sized dHvA orbit regardless of conflicting 2D/3D interpretations.

(2) The authors claim a favorable size comparison of their 2D surface states to the large Cambridge orbits which are interpreted to be 3D, without any additional commentary about the 2D/3D disparity. The obvious lines of reasoning to me are (a) there is meaningful correspondence to the (111) ARPES and dHvA orbit sizes, AND either the ARPES or dHvA orbits are incorrectly claimed to be 2D or 3D, OR there is NO meaningful correspondence to the (111) ARPES and dHvA orbit sizes.

(3) Similarly the authors do not provide any commentary about the fact that the ARPES in-gap states should inherently contribute to transport (for a sample with (111) surfaces), while the lack of SdH oscillations has in part led to the claim of a 3D FS without charge transport. How is a reader supposed to come away with a new insight into the conflicting and controversial dHvA results?

(4) The (111) 2D surface states should not be relevant at all to the dHvA measurements if there are no natural (111) facets in the dHvA crystals.

(5a) The authors claimed agreement in orbit size with the Cambridge group is undercut by the fact that they characterize a semi-minor axis ellipse value of 0.7 to be "close to" 0.49. The authors acknowledge in the same sentence that "These values are not exactly the same ...". Yes they are closer to each other than the small dHvA orbit, but their large >30% quantitative disparity leads to a factor of 2X different orbit frequencies.

(5b) Furthermore it appears the authors make an error of interpreting the Tan et al. ellipse semi-minor axis as a diameter when they compare their value of 0.7 \AA^{-1} (a diameter) to the dHvA value of 0.49 \AA^{-1} (a radius). The Tan et al. supplemental table is for the large alpha orbit with min/max area of $\pi \cdot (0.49)^2 = 0.75 / 1.4 \cdot 0.75 = 1.06$ and a min/max dHvA frequency of $10.47 \cdot A = 8-11 \text{ kT}$. The (111) extremal orbit of their 3D ellipsoid would be intermediate between these values of 8-11 kT. In contrast the ARPES (111) in-gap orbit has an area of $\pi \cdot k_a \cdot k_b = \pi \cdot 0.35 \cdot (1.2 \cdot 0.35) = 0.462$ and $F=4.84 \text{ kT}$. The (111) ARPES value is about 2X smaller than the dHvA alpha-sized orbit.

(6) For completeness of referencing, the two dHvA groups have now published two papers each: Cambridge (Tan Ref. 18, Harstein Ref. 19) and Michigan (Li, Ref. 17 and Xiang PRX 7,031054 (2017)).

I recommend the authors cut the promise of dHvA insight from the abstract and introduction if they cannot properly deliver a message other than "more work would be helpful to solve this complicated problem". They should correct the introductory statements about the different dHvA results (1), and just quantitatively compare ellipse contour sizes without stretching for unjustified statements about agreement.

Also in general, self-grandizing statements about the results "concluding" the discussion on the topological order detract from the impression of the paper. Just an opinion.

(001) ARPES comparison

The Figure 4 schematic presentation of the (001) surface results is confusing in its highlighting of an "umklapp" band interpretation that the authors then claim is incorrect in the text. Maybe the caption should explicitly state that the authors disagree with the umklapp model being shown.

In addition to the disparity of Fermi velocities of the X-point and "umklapp" bands, the authors appear to overlook the obvious deficiency in the circular "shape" of the "umklapp" contour. Two orthogonal overlapping elliptical contour replicas from the X-point arising from separate 2x1 and 1x2 domains of surface reconstruction is the actual prediction by the "umklapp" model and not a circular contour at the zone center. Previous authors [Xu, Ref. 14] did try to draw draw an elliptical 2x1 umklapp contour at the zone center.

While the new interpretation of the 35 eV circular orbit (from ref. 13) as the missing/misidentified third topological surface state on the (001) surface is attractive to claim a unifying interpretation of the different SmB6 surfaces, it still suffers from its complete absence at higher photon energy high symmetry cuts. This suggests a very large bulk-coupling of the in-gap states if they only show up in k-space in the vicinity of the bulk X-points.

First of all, let us express our deepest acknowledgement to all the reviewers. Thanks to their comments, we could improve our manuscript (MS) significantly, providing important additional data, references, and discussion.

In the following part, we made point-to-point responses to each comment.

Reviewer #1 (Remarks to the Author):

1 - Properties of the (111) surface.

As stated in the manuscript the commonly studied (100) surface poses several experimental challenges: Namely the surface reconstruction, the atomic termination and domains and surface polarity [Kim et al., PRB 90, 075131, Zhu et al., PRL 111,216402, Denlinger et al. Arxiv. 1312.6636]. Can the authors comment and describe the expected and obtained structural and electronic properties of the (111) surface in respect to these points? What is the surface termination and is the surface polar? Previous works have demonstrated how surface polarity gives rise to trivial surface states on the (100) surface [Zhu et al., PRL 111,216402] and how seemingly in-gap states arise from surface resonance on different terminations [Hlawenka et al. , Nat. Comm (2015)]. The characteristics and advantages of the (111) surface with respect to these arguments should be discussed and demonstrated in the paper (e.g. DFT, see below).

We agree with the referee that this information is important and was missing in the last version of our MS. Based on the low-energy electron diffraction (LEED) and angle-integrated photoelectron spectroscopy, the current (111) surface would be terminated by Boron clusters. This point has been already published in Physica B 536, 75 (2017) (ref. [25] of the revised MS). We added this information at p. 4 in the revised MS.

From the LEED patterns, we've also found faint facet structures on the (111) surface. However, no influence from them have been found. This point had been already discussed in MS since the previous version (p. 4, the first paragraph of "Surface electronic structure of SmB₆(111)").

About the polarity, the Boron termination could be regarded as surface dipole. However, it has no influence to the topological order, since this dipole appears only on the surface. The bulk SmB₆ lattice has space inversion symmetry, excluding the polarity in the bulk electronic structure, as artificially applied in the slabs in PRL111, 216402 (we referred this article as ref. [35] in the current version). In addition, the "seemingly in-gap" state around the Fermi level shown in PRL111 is clearly different from what we've observed in this work. That state dispersed independently from the bulk bands without connecting insulating bulk balance and conduction bands. Therefore, even if such trivial surface state appeared, it would play no role to determine the topological order of SmB₆. The procedure how to determine the topological order is explained at p. 8 in the revised MS and p. 10-11 in this reply (one of the answers to the reviewer 2).

2 - The labelling of the bands in Fig. 2 and in the text is unclear.

Bands are labelled as S1, S2 and S1' but a clear assignment of these bands is missing and only a guide to the eye is shown in the Fig.2. Can the authors better define the bands they see with photoemission? The bulk band structure of SmB6 is well-known, how do the labelled features relate to it?

The text is very obscure in this regard, I can infer that S2 is the bulk d-f band derived from the hybridization of the Sm 4f and 5d states. S1 is the putative surface topological state as it's dispersing across the gap. The definition of S1' is even more vague and it is later assigned to a Kramer pair of S1, but it rather looks like intensity from the f-band. I encourage the authors to clarify the description of the observed band structure in terms of expected bulk and surface bands.

We agree with the referee that the more detailed ARPES data and analysis to trace the band dispersions around the Fermi level should be provided. For this, we added the ARPES energy distribution curves (EDCs) and momentum distribution curves (MDCs) with its peak positions in Figs. 3 (Γ -M) and S3 (Γ -K).

From the MDCs, two highly dispersive bands S1 and S2 are clearly observed as peaks. From the EDCs, two nearly localized states are observed as indicated by filled triangles in Figs. 3 and S3. In addition, the highly dispersive bands S1 and S2 are observed as broad features on EDCs. While they are difficult to be detected from the single EDC spectrum, comparison with the other spectra indicates that there are finite intensities corresponding to S1 and S2 (see Fig. S4 (a) in the revised SM).

The origin of S1 is the clearest, derived from the surface, because it disperses across the Fermi level while the bulk bands are insulating. S2 is also a surface-derived band. Although S2 lies in the projected bulk bands (The shaded area in Fig. 3 (d) is the projected bulk bands from a theoretical calculation), its dispersion is not influenced by the incident photon energies (Fig. S2 in SM), indicating that S2 is a 2D, surface-related band. Such 2D bands within the projected bulk bands are known as surface resonance and observed on various surfaces. We added a review (ref. [27]) for the information to the readers.

The origin of "F" (we changed its name in the current MS) is more difficult; while it neither shows no 3D dispersion, the localized character of the bulk Sm-4f band justifies such behavior. Therefore, it is difficult to conclude its origin from the spin-integrated ARPES. We discuss this point again in the latter part (reply to the point 4-3, about the interpretation of the SARPES data).

All the discussions about the band assignments are added in p. 4-5, 7 in the revised MS and p. 3-5 in SM together with the relevant Figures (Figs. 3, 4, S2-S4).

3 - The text does not clearly state the bulk or surface nature of the observed band structure. Fig.S2 shows two Fermi surfaces obtained at different photon energy in order to demonstrate the surface nature of the states at the Fermi level. Can the authors provide a more quantitative analysis of the k_z dispersion? On first sight it looks to me like the area of the Fermi surface pockets is different from fig.S2a, S2b and Fig.2b by comparing the size of overlap between neighboring ovals. This is probably just a resolution effect but a more quantitative analysis is needed for a convincing argument. I encourage the authors to show the k_x dispersion as a function of photon energy in the whole 18-35 eV range measured, and to show a comparison between the surface states within the Kondo gap and the bulk states below it.

Finally I think the paper would greatly benefit from a comparison with DFT calculated band structure that would help in the assignment of the bands and with the surface/bulk distinction increasing the value of the paper.

We thank the reviewer for this fruitful suggestion. According to this, we have performed an additional measurement to show the quantitative ARPES peak analysis as a function of incident photon energies. The data and related discussion are shown in Fig. S2 and p. 3-4 in the revised SM. As shown there, S1 and S2 showed no dispersion along the surface normal, together with the incident-photon energies, indicating that they are 2D, surface-derived bands.

As for the DFT calculation, it is quite difficult for us because conventional DFT methods does not work for SmB_6 because of its strong electron correlation. To include such correlation effect and the surface atomic structure (by slab or semi-infinite geometries) is beyond our performance. Instead, we refer to an earlier work based on DFT (ref. [31], PRB93, 195117). Based on the results there, we add the discussions about the qualitative shape and spin polarization of the surface states (p. 7 in the revised MS).

4 - I find the spin-ARPES curves shown in Fig.3 very confusing.

In particular:

-1- The spin EDCs in Fig.3c do show asymmetry for the in-gap state. However there seems to be a significant spin polarization on the valence bands as well (below -0.03 eV). Overall there seem to be a significant asymmetry in all states shown in the EDCs, down to below 0.15eV. The authors should comment on the analysis procedure for the spin ARPES data, in particular how the curves are renormalized, how the polarization is

calculated, and whether the curves shown are raw data or renormalized by the Sherman function (spin-data) error bars should be shown.

We agree with the referee that the detailed procedure to calculate the spin polarization should have been shown in the manuscript (MS). We added it at p. 10 (methods) as:

The spin polarization of the SARPES spectra is calculated by $P = (I_p - I_n)/S_{\text{eff}} (I_p + I_n)$, where P is the spin polarization shown in Fig. 4 (c), I_p (I_n) the raw spectra obtained from the VLEED detector with positive (negative) target magnetization, and S_{eff} the effective Sherman function. S_{eff} is calibrated by the spin polarization of the well-known surface state (Ref. [38] in MS, Rev. Sci. Instrum. 82, 103302). The errors of P is calculated as the standard statistical error, at the same time. Then, the spin-resolved spectrum I_{\pm} , which are shown in Fig. 4, is calculated by
$$I_{\pm} = (I_p + I_n) (1 \pm P)/2.$$

This procedure is the normal way to obtain the spin-resolved spectra. For the SARPES spectra in Fig. 4, no normalization procedure has been applied. The error bars are already shown on the spin-polarization curves.

From these data, thus, one can find that the spin polarization opposite to that of S1 is not an artifact of the analysis but the real signal from the initial electronic states. The origin of them is discussed in the following together with the replies to other questions.

The spin polarization switches at deeper binding energy. In the text this is attributed to the S1' state as "Kramer partner" of S1. This is non-conform with the definition of "Kramer" partner that is found at the time reversal point in the Brillouin zone. If the authors mean the state of opposite spin found at the same k-point (i.e. upper vs lower branches of the topological state) it should be justified how this state can exist seemingly within the valence band. Also, the S2 state at higher binding also shows spin-polarization, however this should be a bulk state (if I understood correctly) and should not be polarized.

As correctly pointed out by the referee, the opposite spin polarization between S1 and "F" is not the exclusive evidence of the Kramers partner. The other possibilities should be discussed. We added this discussion at p. 7. In short:

They fulfill the necessary condition of the Kramers partner, the opposite spin polarizations and apparent dispersion degenerating each other at M. However, the smoking-gun evidence of the Kramers degeneracy is the well-resolved dispersion of two branches in the vicinity of the Kramers point, and such data is not provided in our MS because of the limited energy resolution. Therefore, we just "suggest" that S1 and "F" "could be" the Kramers partner.

The origin of “F” is not clear from spin-integrated ARPES, as discussed on the point 3 by the reviewer, but from the SARPES EDCs (Fig. 4 (c), Cuts 3-4), one can find that there are the spin-polarized peaks corresponding to “F” which are well separated from S1 and have the opposite sign to S1. From this data, we suppose that a part of “F” is also derived from the surface electronic structure (it is explicitly stated that it is just a supposition). Note that it does not exclude possible bulk 4f bands overlapping “F”.

About the spin polarization of S2, we’d like to point out again that S2 is not bulk but surface band. Therefore, the spin polarization of S2 is not surprising, as a surface band. These discussions are added in the revised MS at p. 5 and 7.

-2- From fig.3e, the in-plane spin polarization appears to be chiral and tangential around the pockets. However, in SM figS3 it appears that the spin polarization along Gamma-M is along the [-1 -1 2] direction, while it should be towards [-110] according to the main text. Is this a mistake in the legend?

The referee is right. It is our mistake. We thank the referee for the correction.

This point is corrected in the revised SM (Fig. S6 (a)).

-3- The out of plane polarization shows indeed a strong asymmetry along Gamma-K in Fig.3b, but its spin texture is not clear from the text and Fig.3e. Although an out of plane polarization is not symmetry forbidden along Gamma-K, in the simplest terms I would expect that the out of plane polarization cancels out along this direction because the two oval pockets are degenerate and should carry opposite spin polarization. How is the out of plane signal justified? Is it an effect of photoemission cross section? Can the authors elaborate on this and show it more clearly in the Fig.3e?

We added an explanation about this point in the SM (p. 8-9). As shown in Fig. R1 below (the same figure is shown as Fig. S7 in SM), the surface symmetry operations DO NOT cancel the out-of-plane spin polarizations along Γ -K.

Figure R1 Schematic drawings of the spin-polarized Fermi contours (FCs) with symmetry operations. (a) Single FC at a M point. (b) FCs multiplied by 3-fold rotation. (c) FCs multiplied by time inversion. (d) FCs multiplied by the combination between (b) and (c).

Overall the spin-ARPES analysis seem shortcoming and does need a more thorough discussion. I recommend the authors to review the spin-ARPES analysis commenting also on possible spurious effects of the technique, as shown in previous works [Heinzmann et al. J. phys. Cond. Matt. 24, 173001 , Jozwiak et al., Phys. Rev. B 84, 165113, S. McKeown Walker et al., Phys. Rev. B 93, 245143]

Thanks to the comments by the reviewer 1, we could improve the explanations and discussions in the MS thoroughly, making the MS more convincing and comprehensive.

Some artificial spin polarization of photoelectrons, as discussed in the papers the reviewer proposed could also occur in our SARPES spectra. However, we'd like to point out that the spin polarization whose sign inverts with respect to time inversion, as shown in Fig. 4, cannot appear by such artificial spin polarizations.

Although it is sometimes shortcoming to connect the observed spin polarization of the photoelectrons to those of the initial states directly, it is evident that the initial states, S1, S2 and F in this MS, is somehow spin-polarized and its sign inverts according to the surface symmetry. This information is enough to discuss the topological order of the sample from its

surface states. In MS, we carefully state the ARPES data to compare the theoretical results (p. 7-8) added in the current version.

5 - I am a bit puzzled by the discussion paragraph: The authors refer to the paper { Hlawenka et al , Nat. Comm (2015) , ref.13 in manuscript } and write that the “slope of the “umklapp” band is different from that of its counterpart”. This does not seem obvious to me from the referenced paper and the authors should better/quantitatively justify this claim.

We found the different slopes between the surface and “ Γ -state” band (“umklapp” band in the previous MS. We changed its name for better readability) from Fig. 2 (c) in ref. [14] (ref. [16] in SM, Hlawenka et al., Nat. Comm 2018). To show it clearly, we made Fig. R2. This point is explained explicitly in the revised SM (p. 10).

Fig. R2 A copy from Fig. 2 (c) of ref. [14] (Hlawenka et al., Nat. Comm 2018). We added a green (yellow) line to guide the dispersion of the surface (“ Γ -state”) band. A dashed line is the mirrored dispersion of the surface band for the sake of comparison.

Finally throughout the paper the authors state very strongly that their findings are a conclusive prove of the topological nature of SmB6, I think even if all my previous points are appropriately addressed this claim is too strong as the authors themselves discuss in the last paragraph stating that “the current ARPES result shows agreement and disagreement with the two conflicting report” and “comparison between the ARPES data from multiple surface orientation” ... “and other bulk sensitive methods would be helpful in solving this complicated problem”

The topological order of the material is reflected to any orientations of surface electronic structures, at least for the strong topological insulators, which is the current case. Therefore, in order to determine the topological order, to observe the surface electronic structure and its topology from one surface is enough. The difficulty of the $\text{SmB}_6(001)$ surface as discussed in the manuscript is now solved by using the (111) surface. All the problems of the analysis and interpretations pointed out by the reviewers are now solved. Therefore, we are quite sure that the topological order of SmB_6 is unambiguously determined. We are reluctant to weaken the scientifically valid statement without any reasonable weak points.

All the phrases the reviewer referred here is from the discussion part of the previous MS to compare the ARPES/SARPES results with the dHvA measurements. From this aspect, actually, there are both agreement and disagreements and further studies in addition to the current data is desirable. However, this subject is independent from the topological order of SmB_6 . For the information, the comparison with the dHvA results are significantly revised according to the suggestions by the reviewer 3.

6 - Overall, I find the paper lacking in overall presentation and both readability and figures should be improved for publication in Nat. Comm.

We thank the reviewer 1 again for his/her questions and suggestions. Based on them together with those from the other reviewers, we believe that the revised MS and supplementary material (SM) is now improved significantly and meets both readability and validity for publication in Nat. Comm..

Reviewer #2 (Remarks to the Author):

The so-far debatable ARPES data consisted of electronic states at X_{bar} that were reminiscent of the bulk Fermi surface contours (bulk contours = ellipsoids at bulk X points). These X_{bar} states were however shown to be spin-polarized [Xu ncomms 2014] favoring the interpretation of topological states. In this sense, the new data by Ohtsubo et al. does not give new information even if it was acquired on a new surface. To make my point clear: the bulk ellipsoids at X when projected on the (111) surface Brillouin zone would exactly give the contours seen by Ohtsubo and coworkers. The situation is reminiscent of the 2DEG observed in bare surfaces of transition metal oxides. For example, the 2DEG observed on different surfaces of SrTiO₃ -that is in fact a confined version of the bulk conduction band- is projected on the (001) surface [Santander Nature 2011], the (110) surface [Rodel PR applied 2014] and the (111) surface [Walker PRL 2014] giving each time different contours which are in perfect agreement with the corresponding projection of the very same bulk state. Why don't the authors consider this possibility for SmB₆? A counterargument would be the observed spin polarization but the existence of spin polarization already known since 2014 for the very same states on the (001) surface of SmB₆.

The problem we solved in this MS is how to determine the topological order of SmB₆ from its surface electronic state obtained by ARPES and SARPES.

In order to determine the topological order, one has to obey the three steps:

- a) Detect the surface bands dispersing continuously across the Fermi level. And count the number of closed Fermi contours (FCs) enclosing surface time-reversal-invariant momenta (TRIM).
- b) Observe them by SARPES (or other methods sensitive to spin-orbital polarization of the surface states) to check if they are spin polarized or not. The number counted in (a) has to be doubled for the degenerate states.
- c) Count the total number of FCs. If it is an odd number (1, 3, 5, ...), then the sample is a (strong) topological insulator. If even (0, 2, 4, ...), the sample is normal, trivial insulator. Although the X-bar states on SmB₆(001) and its spin polarization is well known, it is not enough to conclude the topological order of SmB₆, since one has to check the other surface states around the other surface TRIMs. This point, the existence or absence of the additional surface states around Gamma, is what is discussed in ref. [14] (Hlawenka et al., Nat. Comm 2018) and has been a long-debating problem.

On the (111) surface, this problem can be avoided, since there are only two inequivalent surface TRIMs. Although the M-bar surface state found in this work is similar to the X-

bar ones on the (001) surface, the important point is that the M-bar state on (111) is the unique metallic surface state and thus we could determine the topological order of SmB₆ without any ambiguity.

The similar appearance of the M-bar state on (111) and the X-bar on (001) would suggest the common origin of them. But it does not play any role to consider the topological order, as discussed above. We agree with the reviewer that the microscopic origin of them might also be an interesting subject, but we think this subject should be discussed elsewhere to focus on the main theme of this MS, the topological order of SmB₆.

This discussion had been shown partially in the previous MS (“A (111) surface of SmB₆”). We revised it to provide enough information how to determine the topological order from ARPES and SARPES results to general readers. The revised discussion is shown in p. 8 in the revised MS.

Concerning the SARPES data: The observation of spin polarization is very important but SARPES data has an energy resolution (30 meV) that is equal to the total range of the energy dispersion of the S1 state. I am aware that this value is very competitive in terms of resolution for SARPES but in my opinion the small energy range of dispersion for the near-EF states in SmB₆ makes SARPES unsuitable for this specific system.

The energy window of SARPES in this work, 30 meV, is nearly the same size to the observed dispersion of S1, as the referee correctly pointed out. But we do not think it is the severe problem to pick up the spin polarization information of S1 from SARPES. Such problem occurs only if the energy window is much wider than the total dispersion and some additional features included there. It is not the case of ours. The S1 band is observed as the single spin-polarized peaks by MDCs and EDCs and well resolved from the lower ones (S2 and “F”, we changed its name from S1’). This discussion is added to p. 6-7 in the revised MS.

I also have some specific comments on these data:

- In Fig. 2d, cuts 1 and 2 do not seem to go through the S1 state. Yet, the spin polarization is clear. How can the authors explain that? Could it be that the S1prime is polarized INSTEAD of S1?

It indicates that S1 disperses as guided in the revised Fig. 3 (a) and 3 (d); at 0.02-0.03 eV at Cuts 1-2. Note that the spin polarizations (negative for +k_y) are the same for Cuts 1-4.

The spin polarization of “F” is the opposite to those of S1 as shown in Cuts 3-4. At Cuts 1 and 2, the polarization from “F” is not clear. It might be masked by the too strong spin polarization of S1.

- The authors state that S1prime and S2 have opposite polarization to S1. How can this be reconciled with what we know about SmB6? S1prime is a flat band of f-origin while S2 is the highly dispersing d-band. These features are well known and spin polarization is not expected for neither of the two.

At first, we'd like to point out that S2 is not bulk but surface band. Although S2 lies in the projected bulk bands (see the new Figs. 3 and S3), its dispersion is not influenced by the incident photon energies (Fig. S2 in the revised SM), indicating that S2 is a 2D, surface-related band. Such 2D bands within the projected bulk bands are known as surface resonance and observed on various surfaces (for example, see ref. [27]). Therefore, the spin polarization of S2 is not surprising, as a surface band.

The origin of “F” (S1' in the previous version) is more difficult. While it neither shows no 3D dispersion, the localized character of the bulk Sm-4f band justifies such behavior. From its spin polarization, we supposed it to be surface-derived one, but it is explicitly stated as a supposition (p. 7 in the revised MS). In addition, we'd like to stress that we do not exclude possible bulk 4f bands overlapping “F”.

To conclude: Although the data is interesting and its acquisition must have been very challenging I cannot recommend publication because the authors are guiding the reader towards the conclusion of topological states with no new insight with respect to what we so far knew (i.e. the FS contours are new but well expected and the spin polarization had been already observed). I am not suggesting a bare reformulation of 2-3 sentences in the manuscript but a fundamental rewriting of the discussion -and not only- to include all different possibilities (confined bulk states vs. topological states) and to mention the limitations of SARPES data. I believe that the reader should have the right to choose which interpretation is the best without being influenced. In contrast to the authors' statement, in my view, the “second level” of the debate on SmB6 is far from being concluded.

We thank the reviewer 2 again for his/her questions and suggestions. Based on them together with those from the other reviewers, we believe that the revised MS and supplementary material (SM) is now improved. In particular, we hope that the additional

discussion, how to determine the topological order by ARPES and SARPES data, convinces the reviewer 2 that our result provide the new and inevitable information to determine the topological order of SmB₆.

Reviewer #3 (Remarks to the Author):

In-gap Fermi surface size

The most striking and unexpected experimental result concerns the large size of the in-gap state elliptical contours in comparison to the in-gap states for the (100) and (110) surfaces. For instance Ref. 21 provides some size numbers in terms of dHvA orbit frequencies (discussed more later): For the (100) surface the in-gap state orbit size is $F=3.5$ kT and for the (110) surface the smaller in-gap contour is 2.0 kT. In contrast, here for the (111) surface $k_a=0.35$ and $k_b=1.2*0.35=0.42$, the dHvA frequency is $F=10.47*Area = 10.47*\pi*0.35*0.42 = 4.84$ kT. Also for comparison Ref. 21 gives the Sm 2.5+ mixed valent and LaB6 trivalent bulk 3D ellipsoid extremal orbit sizes of 4.5-6.0 kT and 7.5-10.0 kT, respectively.

To better understand this large in-gap state contour result, it would be instructive if the authors provided supplementary information comparing the in-gap states to the bulk d-states. I.e. (a) present and quantify a constant energy map from the deeper binding energy bulk d-band just below the highest f-state (approx. -50 meV), and (b) quantify the kF-shift between the in-gap states and the extrapolation of the d-band states to EF.

For (a) does it have an even larger overlap of elliptical contours? Also do the d-states outside the gap exhibit any 3D variation in size between 19, 26 and 35 eV? For (b) the (100) surface numbers provided by Ref. 21 indicate a kF-shift as large as 0.11-0.13 (inverse angstroms). Here visually from Fig. 2(c) the extrapolation of the d-band dispersion appears to have a smaller kF-shift of about 0.08.

We thank the reviewer for such an interesting suggestion. We've made the proposed analysis as shown in the revised SM (p. 5-7, Fig. S5 and Table I). Actually, the contour from the analysis (a) resulted in the even larger than the actual Fermi contour. The analysis (b) showed the kF-shift of 0.08, as the reviewer correctly expected.

Providing such quantitative numbers allows the authors or readers to then contemplate some deeper meaning to the difference with the (100) surface. Is it just a directional projection effect, or does it reflect a difference in a Kondo surface breakdown scenario as proposed by Alexandrov PRL 114, 177202 (2015) and Peters, Ref. 4 or something else? E.g. comparison of a 2D state to the projection of bulk 3D ellipsoid to the surface can also be tricky. For instance Koitzsch et al. Nat. Comm. 7, 10876 (2016) measure bulk-sensitive (111) FS contours of trivalent CeB6(111) and find the contours to just touch each other consistent with the 3D ellipsoids and NOT overlap as might expect projected to the

surface. In this illusory sense, the 2D in-gap state of the 2.5+ valence system here is even larger than the trivalent CeB₆ FS contour.

According to the suggestion from the reviewer, we made a comparison of the current data with the known trivalent CeB₆ FS contour. This analysis is shown in p. 5-6 in the revised SM.

In parallel, we added a discussion about the Kondo breakdown proposed by Alexandrov (this work is now referred as ref. [4] in the main text). This discussion is added in p. 9 in the revised MS.

Overlap of topological states

Is the overlapping of topological surface states from separate neighboring TRIM points a new unprecedented result not reported earlier in the literature? The authors should comment on this and discuss. Certainly there are reported complex interaction effects of the dual Dirac cones when two bulk TRIM points project onto the same surface BZ point, but that is a different scenario than the result here.

We also appreciate the reviewer to point out this interesting viewpoint. The overlap of topological states has been discussed in topological crystalline insulators (TCIs) (reference [5] in SM) and a Lifshitz transition is expected to appear in the overlapping region of the surface Dirac cones.

In our data, the overlapping region is shown along the k_F points along Gamma-K (Fig. S3, ~ 0.3 and 0.7 \AA^{-1}). However, further analysis about expected critical points derived from the overlapping topological states is impossible because of the limited energy resolution (15 meV). To obtain further insight into this point, the better energy resolution is desirable. This point is added as a future perspective of this work at p. 6-7 in the revised SM.

Does one expect any interaction between the overlapping surface states? If so, one would imagine that surface state electron orbits might not be broken down into smaller sized orbits of the overlap region and the central warped hexagon. What dHvA frequency corresponds to these two smaller orbits? Or can it be argued that there should be no interaction, and that since the in-gap states are single-spin states are they allowed as Fermions to share the same momentum point if their spin states are orthogonal?

Within the ARPES energy resolution in this work, we found any signature from the Fermi contours to break down to the smaller-sized contours. However, it does not exclude the

possible separation of the FCs with the energy and momentum separation smaller than the experimental resolution. We estimated the dHvA frequencies corresponding to the smaller-sized FCs: warped hexagon and 6 shallow ovals and made a discussion in p. 7 and Table I in the revised SM.

Spin-ARPES

(1) A crucial citation that is missing is Baruselli and Vojta, PRB 93, 195117 (2016) which provides a theoretical prediction for the spin texture of the topological states for the (100), (110) and (111) surfaces of SmB₆. In particular they predict opposite clockwise/counterclockwise in-plane helicities of the spin for different winding numbers ($w=\pm 1$) for all the surfaces as well as a specific variation of the (111) out-of-plane component of the spin for the case of $w=+1$. The authors here also have in-plane and out-of-plane results and should compare and discuss their measured spin texture to that prediction.

The schematic of the (111) spin-ARPES results in Fig. shows a clockwise spin-helicity that would correspond to the Baruselli parameter $w=-1$. This appears to agree with the claim that the (001) spin-ARPES result also agrees with $w=-1$.

We thank the referee to introduce this important work. It is referred as ref. [31] in the revised MS and we added a discussion at p. 7-8 to introduce its agreement with the observed spin-polarized Fermi contours. In this discussion, the possible difference between spin polarization of photoelectrons and those of the initial states are carefully discussed.

(2) The authors use two different geometries to measure the in-plane spin-polarization at the ellipse tips. In Fig. 3(a,c,d) the tilt angle is used to access $k_y = \pm k_F$ and the spin-polarization is observed towards the (-110)/(1-10) directions which is tangential to the ellipse tip. In the supplemental Fig. S3(a), the sample azimuth is rotated and the polar angle is used to access $k_x = \pm k_F$ at the ellipse tips and strong spin-polarization is observed towards the (-1-12) or (11-2) horizontal directions. However, due to the azimuth rotation, this direction is now “radial” to the ellipse. How am I supposed to understand both tangential AND radial in-plane spin-polarization component results between Figures 3(a,c,d) and S3(a)?

It was our mistake. We are sorry for the confusion of the referee. The spin polarization observed in Fig. S3 (Fig. S6 in the current version) is along $(-110)/(1-10)$. This point is corrected in the revised SM.

(3) Another obvious citation that is missing is another spin-resolved ARPES measurement of SmB₆ by Suga et al. JPSJ 83, 014705 (2014). In that study, they failed to reproduce the spin-polarization of the (001) surface in-gap states reported by Xu et al. [Ref 12], but along the way they did report strong spin-polarization of the SmB₆ f-states using 35 eV circular polarized excitation (and none for linear polarization). I see that the spin-ARPES measurements here are for linearly polarized 26 eV photons. I believe it is useful for the authors to cite this other work and comment on how or how much they are able to avoid Sm 4f-state spin-polarization contributions?

We thank the referee again for introducing an important work. We added this to the reference (ref. [39]) and made a short statement at p. 10, presenting that such artificial spin polarization should not appear because we used linearly polarized photons.

(4) While the authors characterize spin-ARPES results as “regarded as indisputable evidence of the topological order”, in fact there exists an example of spin-ARPES claims of topological order in surface states of another hexaboride, YbB₆ (Xu, arXiv:1405.0165), that was in fact disputed (Kang, PRL 116, 116401 (2016)).

We agree with the referee that the spin polarization of single surface state is not enough to determine the topological order. In order to determine the topological order, one has to obey the three steps:

- a) Detect the surface bands dispersing continuously across the Fermi level. And count the number of closed Fermi contours (FCs) enclosing surface time-reversal-invariant momenta (TRIM).
- b) Observe them by SARPES (or other methods sensitive to spin-orbital polarization of the surface states) to check if they are spin polarized or not. The number counted in (a) has to be doubled for the degenerate states.
- c) Count the total number of FCs. If it is an odd number (1, 3, 5, ...), then the sample is a (strong) topological insulator. If even (0, 2, 4, ...), the sample is normal, trivial insulator.

For YbB₆, the spin polarization is observed only for one band around X but no spin information about the other bands are provided in arXiv:1405.0165.

In our case, $\text{SmB}_6(111)$, S_1 is the unique surface state dispersing across the bulk gap. It merges into projected bulk bands and clearly spin polarized. Therefore, there is no ambiguity about the topological order of SmB_6 .

This discussion is added in p. 8 in the revised MS.

I do see words in this paper about being able to eliminate geometric contributions to the spin-polarization. Do the authors have a particular citation that describes the potential instrumental/geometrical artifacts in the spin-ARPES technique?

In general, for the photoelectrons emitted along the polar angle, the positive and negative angles are not exactly equivalent. For example, the photon incidence itself breaks makes the positive/negative emission angles inequivalent. In contrast, along the tilt angle, the geometric condition for photoemission is completely equivalent for positive/negative emission angles even if the photon incidence is included. Such effect can be calculated as the different transition matrix element for theoretical model including the photoexcitation process. We added an explanation in p. 2 of SM and a typical example as ref. [2] there.

Relation to dHvA controversy

The abstract and introduction promise new insight to the dHvA controversy. It was then disappointing to find that this boiled down to a very incomplete (and also incorrect) comparison of the size of the large ARPES in-gap state elliptical Fermi contour to the large alpha-band trivalent LaB_6 -like dHvA orbits reported by the Cambridge group (Ref. 18,19). The authors note that (a) their (111) 2D surface state orbit size is much larger than the small orbit size by the Michigan group (claimed to be 2D and originating from (110) surfaces), and (b) that the 2D ARPES contours have semi-agreement to the large Cambridge orbits (which are claimed to be 3D). There is much to be discussed about problems with the presentation of their comparison to dHvA:

(1) The introduction statement “ref. [17] reported small 2D shapes whereas ref [18, 19] reported large 3D shapes” is incorrectly stated. Both dHvA groups agree experimentally in the presence of a strong amplitude small-sized orbits (with $F_{\text{min}}=290\text{T}$), but differ in their interpretation of 2D origin (Michigan) versus 3D origin (Cambridge, $F_{\text{max}}=700\text{T}$). Ref. [18,19] “additionally” report multiple higher frequency weak-amplitude orbits greater than 1 kT including trivalent-like 10 kT sized 3D orbits.

Hence the ARPES large ellipse orbit size disagrees with (or has no correspondence) to the main quantum oscillation result of “both” dHvA groups. As stated earlier, the

authors could provide a quantitative comparison of a potential smaller ARPES overlap orbit to the strong small-sized dHvA orbit regardless of conflicting 2D/3D interpretations.

We thank the reviewer for the correction and kind instruction from his/her expertise. It is quite helpful for us to improve MS and make better explanation about the correspondence between the current ARPES/SARPES results and known dHvA results.

According to this comment, we corrected the introduction of the earlier results (p. 2) as: All the results refs. [18-21] reported the small-sized orbits lying at the Fermi level, but its interpretation is still under debate: 2D [18, 21] and 3D [19, 20].

(2) The authors claim a favorable size comparison of their 2D surface states to the large Cambridge orbits which are interpreted to be 3D, without any additional commentary about the 2D/3D disparity. The obvious lines of reasoning to me are (a) there is meaningful correspondence to the (111) ARPES and dHvA orbit sizes, AND either the ARPES or dHvA orbits are incorrectly claimed to be 2D or 3D, OR there is NO meaningful correspondence to the (111) ARPES and dHvA orbit sizes.

(3) Similarly the authors do not provide any commentary about the fact that the ARPES in-gap states should inherently contribute to transport (for a sample with (111) surfaces), while the lack of SdH oscillations has in part led to the claim of a 3D FS without charge transport. How is a reader supposed to come away with a new insight into the conflicting and controversial dHvA results?

We agree with the reviewer that the apparent correspondence between the 3D, charge-free Fermi surface from dHvA and 2D, probably conducting FS by ARPES was too much simplified in the previous MS.

According to the suggestion from the reviewer here and the point (1)-(3) above, we discard our previous claim about the correspondence of the FC sizes obtained from ARPES and dHvA (Cambridge).

(4) The (111) 2D surface states should not be relevant at all to the dHvA measurements if there are no natural (111) facets in the dHvA crystals.

We thank the reviewer 3 again for the instructive comment. Based on this, we provide the size of the 2D surface FCs as a reference information for the sake of comparison with the future dHvA works on the (111) face of single crystal SmB_6 .

This point is shown in p. 7 and Table I of SM.

(5a) The authors claimed agreement in orbit size with the Cambridge group is undercut by the fact that they characterize a semi-minor axis ellipse value of 0.7 to be “close to” 0.49. The authors acknowledge in the same sentence that “These values are not exactly the same ...”. Yes they are closer to each other than the small dHvA orbit, but their large >30% quantitative disparity leads to a factor of 2X different orbit frequencies.

(5b) Furthermore it appears the authors make an error of interpreting the Tan et al. ellipse semi-minor axis as a diameter when they compare their value of 0.7 \AA^{-1} (a diameter) to the dHvA value of 0.49 \AA^{-1} (a radius). The Tan et al. supplemental table is for the large alpha orbit with min/max area of $\pi*(0.49)^2 = 0.75 / 1.4*0.75 = 1.06$ and a min/max dHva frequency of $10.47*A = 8-11 \text{ kT}$. The (111) extremal orbit of their 3D ellipsoid would be intermediate between these values of 8-11 kT. In contrast the ARPES (111) in-gap orbit has an area of $\pi*ka*kb = \pi*0.35*(1.2*0.35) = 0.462$ and $F=4.84 \text{ kT}$. The (111) ARPES value is about 2X smaller than the dHvA alpha-sized orbit.

We thank the reviewer again for the correction. We removed the underestimated size of the 3D Fermi surface from dHvA.

As stated above, we've discarded most of the claims about the correspondence between ARPES and dHvA in the revised MS.

(6) For completeness of referencing, the two dHvA groups have now published two papers each: Cambridge (Tan Ref. 18, Harstein Ref. 19) and Michigan (Li, Ref. 17 and Xiang PRX 7,031054 (2017)).

Thank you for the introduction. The reference (Xiang PRX 7) is added to the revised MS (ref. 21).

I recommend the authors cut the promise of dHvA insight from the abstract and introduction if they cannot properly deliver a message other than “more work would be helpful to solve this complicated problem”. They should correct the introductory statements about the different dHvA results (1), and just quantitatively compare ellipse contour sizes without stretching for unjustified statements about agreement.

According to the advice from the reviewer 3, we removed the statement about dHvA from the abstract and discarded some claims about the correspondence between 2D FCs

obtained by ARPES in this work and 3D ones from dHvA. In the revised SM (moved from the main text), the sizes of the 2D FCs from ARPES (large ovals and smaller, broken-down ones according to the suggestion by the reviewer 3) are provided as a reference information for future dHvA works.

Also in general, self-grandizing statements about the results “concluding” the discussion on the topological order detract from the impression of the paper. Just an opinion.

The procedure to determine the topological order of the materials by ARPES and SARPES is well established and we’ve succeeded to apply it to SmB₆ without any ambiguity, as discussed above (p. 18-19, p. 10-11 (reply to the reviewer 2) and p. 8 in the revised MS). Such success has been achieved for the first time. Therefore, we don’t think our remark, “concluding the discussion on the topological order”, is a self-grandizing one. We are reluctant to weaken the scientifically valid statement without any reasonable weakpoints. Although our interpretation of the earlier dHvA works had many errors and uncertainties as the reviewer 3 correctly pointed out, they are not directly related to the topological order of SmB₆ determined by ARPES and SARPES.

(001) ARPES comparison

The Figure 4 schematic presentation of the (001) surface results is confusing in its highlighting of an “umklapp” band interpretation that the authors then claim is incorrect in the text. Maybe the caption should explicitly state that the authors disagree with the umklapp model being shown.

We thank the reviewer for the good suggestion. We agree with it and changed the band notation from “umklapp” to “ Γ -state”.

In addition to the disparity of Fermi velocities of the X-point and “umklapp” bands, the authors appear to overlook the obvious deficiency in the circular “shape” of the “umklapp” contour. Two orthogonal overlapping elliptical contour replicas from the X-point arising from separate 2x1 and 1x2 domains of surface reconstruction is the actual prediction by the “umklapp” model and not a circular contour at the zone center. Previous authors [Xu, Ref. 14] did try to draw draw an elliptical 2x1 umklapp contour at the zone center.

Although Xu has predicted the elliptical FC around Gamma, it has not been clearly observed by ARPES until Hlawenka ref. [14] did (Fig. R3; it is copied from Fig. 1C of

ref. 14). The “Umklapp” one is the “ Γ -state” band in our revised MS and shows clearly circular, not elliptical, shape of FC. It is why we put the circular, single FS in Fig. S8 (b). (This discussion is moved from main text to SM).

Fig. R3: Fermi contour observed on SmB₆(001) by Hlawenka et al. (ref. 14). This figure is copied from Fig. 1C of ref. 14.

While the new interpretation of the 35 eV circular orbit (from ref. 13) as the missing/misidentified third topological surface state on the (001) surface is attractive to claim a unifying interpretation of the different SmB₆ surfaces, it still suffers from its complete absence at higher photon energy high symmetry cuts. This suggests a very large bulk-coupling of the in-gap states if they only show up in k-space in the vicinity of the bulk X-points.

We intended to propose a new interpretation, the circular FS at Gamma on (001) to be the other surface state, just as a hypothesis to explain the apparently conflicting ARPES result from SmB₆(001). The main data in this MS is the FCs from the (111) surface.

To make this point clearer, we moved this discussion to SM and added some sentences at p. 10 of SM:

It should be noted that this interpretation is just a possibility based on known results. The detailed origin of the “ Γ -state” band would be elucidated by future ARPES and SARPES works with varying incident photon energies and polarizations.

We thank the reviewer 3 again for his/her questions and suggestions. Based on them together with those from the other reviewers, we believe that the revised MS and supplementary material (SM) is now improved significantly and meets both readability and validity for publication in Nat. Comm..

Reviewers' comments:

Reviewer #1 (Remarks to the Author):

The authors have answered, to some extent, my questions and have improved the manuscript. The characterization of the (111) surface state is now more exhaustive, the bands are now clearly labelled and better quantified through MDC-EDC analysis; also, the out of plane spin polarization is now clear and the readability overall has improved, even if it is still not perfect.

However, there are still important shortcomings in this version of the manuscript and I cannot recommend publication in Nature Communications at this stage. My main concerns are related to the analysis/interpretation of the data presented and the strong conclusions that are drawn from it.

My biggest concerns are the following:

1 - Bulk vs surface band structure:

The definition of S2 and F as surface states is still not convincing to me. In the SM the authors now show the photon energy dependence of two MDCs, crossing S1 and crossing S2 in fig. S2(b) and (c) respectively.

The photon energy range spanned in this experiment covers 80% of the Gamma-R distance as discussed by the authors. However the initial photon energy of 15 eV does likely not correspond to the actual bulk Gamma point, and the range is more similar to 50-60% of Gamma-R and 30-20% of R-Gamma (which carries the same information of Gamma-R because of inversion symmetry). More accurately the energy range used spans 40% of the bulk Brillouin zone.

At the same time, while the MDCs for S1, I agree, are more likely to indicate a two dimensional feature, the same cannot be said for S2, where there is a visible shift of the peaks in k_y for curves at, for example, 15,19,20,21,32 eV with respect to the peaks well centred on the green mark at different photon energy. Also, in the curves at 22-24 eV the peaks are not visible, but it looks like they could just be at the edge of the shorter k -range shown. This is likely just some strong background non-uniformity and the peak could not be visible because of the cross section, but the data quality here presented in this k_z range studied is insufficient to conclude that S2 is a surface state.

An additional question arises, if these (S2, F, S1) are all surface states why the bulk Sm d- and f- states are not visible in the ARPES spectrum? These are usually very visible in SmB6 at these photon energies. S2 also seems to continue at deeper binding energy, below 0.2 eV (Fig.S2(a)), is this also a surface resonance?

In summary: While it is proven that, on the (001) surface, there can be surface resonances, the arguments for lack of 3D dispersion in S2 is weak in this case because of the limited k_z range spanned, and evidence of MDC peaks centered at different k_y for different photon energies. Furthermore, the absence of 3D bulk bands and the similarity of S2 and F with the respective Sm d- and f- states is suspicious.

For a reader looking at this data, as presented here, a more likely scenario would be that these are the standard d and f bulk states that one would find projecting onto the (111) surface the bulk bands. Be noted that this scenario would of course not affect the possible presence of topological states at the surface.

2 - Spin-ARPES Analysis:

I am still puzzled by the spin-ARPES data and their interpretation:

It must be noted that acquiring spin-ARPES data within the small gap of SmB6 is a very challenging task, so the data quality (signal to noise ratio) here presented, is notable. However, the spin profiles, here derived from the raw data as described in the methods, are particularly odd:

Fig 4 (a-b) shows the spin-MDC at the Fermi level. I find the shape and position of the peaks, related to the putative topological state, a matter of concern. In panel (a) the left peak (negative k) only shows spin up (nominally, red) component, yielding a $\sim 100\%$ spin polarization; the right

peak has both spin up and down components. This is not necessarily concerning because states at opposite momenta could have quantitatively different spin polarization (i.e. [Zhu et al, PRL 112,076802 (2014)]), but here the blue and red peaks have a different shape, in particular the full width half maximum is different as well as the peak location. This is even more evident in Fig 4.(b): here the blue peak at negative k seem to have different peak location respect to the red peak (shifted to higher k). In the positive k side this is also very obvious, the blue profile shows a peak around 0.35 \AA^{-1} , but it is impossible to define an actual peak position for the red curve. The k range shown is too limited, and an actual peak shape cannot be distinguished. The spin up and down distribution curves should show the same peak shape if they are referred to the same state, because the additional spin selectivity of the technique does not change the spectral function of the state, but merely the intensity. the apparent shift in peak location between opposite spin signals is more indicative of a two components feature (one spin up and one spin down). This scenario is also corroborated by the crossing between red and blue curves at the right edge of Fig. 4(b), but the k range of fig.4(b) is too small and should be extended to provide any meaningful information on the right peak.

Secondarily, the authors have stated in the rebuttal that the spin-ARPES signal does not undergo any renormalization. Does that imply that raw data curves are not renormalized with respect to detector efficiency (namely, in the VLEED, the different scattering efficiency of the target for up and down magnetization)? Does that mean the detector is perfectly equally efficient for all opposite spins? This question arises looking at the EDCs. It is obvious that there is some polarization somewhere but, as stated by the authors, all bands in these plots seem to be spin polarized. This is surprising, also in view of the discussion about bulk vs surface states. Generally, the spin-ARPES normalization process can be quite complicated and it benefits from the presence of spin degenerate states that show no polarization or a clear opposite spin band as comparison. I encourage the authors to present the spin-ARPES data more clearly with, if needed, ulterior analysis. The k-energy range should be large enough to fully distinguish the peaks considered and, if the opposite spin signals show non-equivalent peak shapes, this should be justified. A spin-degenerate background or state would also make the analysis more convincing.

3 - Conclusions

I renew my scepticism when the authors state that their work "conclude the discussion on the topological order in SmB6". This is not simply because of the band assignment and spin-ARPES data as previously discussed.

Even if these points are clarified, Spin-/ARPES is usually insufficient to conclusively define a topological insulator phase. ARPES may lack the ability of detecting some electronic states for several reasons (resolution, range, matrix elements). For this reason, while ARPES is a great tool to provide evidences of a topological insulator phase, and has been successfully employed in many cases, it will likely not provide a definitive answer, if further techniques are not employed. This is particularly difficult for a complicated system like SmB6, where the correlation gap is very small, and the samples are well known to be unreliable. A great number of contrasting results have been shown throughout the years with evidence both in favor and against the topological picture [Xu et al. J. Phys.: Condens. Matter 28,363001(2016)], and this study, while interesting and important, could not fundamentally provide a definite conclusion to the SmB6 case without a systematic study of different samples and with different techniques.

As I have previously stated I think that the study of the (111) surface of SmB6 is very interesting, important and timely and can contribute to the heated discussion on the topological nature of the material. While the data could be worth publication, the present manuscript still has some critical shortcomings.

Reviewer #2 (Remarks to the Author):

I have read the revised manuscript by Ohtsubo et al., all referee's comments and the authors'

rebuttal. The new manuscript has no fundamental changes with respect to its previous version and the authors stay very strongly in their position that their findings are a conclusive proof for the topological nature of SmB6.

I disagree. I find the association of all bands to surface states very hasty. Reviewer 1 and myself expressed our concerns about that. The authors stay in their position that the lack of k_z dispersion for S1 and S2 is the underlying proof for their surface origin even if –at least S2- has been attributed to the bulk 5d states of Sm in many previous studies and has counterparts in all other hexaborides. As for attributing S1' (or F) to a surface band because of the observation of spin polarized peaks, this is not a convincing argument knowing that the Sm 4f bulk states should be exactly at the corresponding E-k position. Lack of k_z dispersion has been observed for states that are well known not to be new surface states. This can be due to the fundamental limitation of ARPES in resolving momentum components along the surface normal (see Strocov 2003), or because the states in question are a confined version of bulk bands (e.g. quantum well states). This is exactly what happens in 2DEGs on the surface of perovskite oxides. Such a scenario has been considered by Hlawenka et al. when they discussed a “surface-localized” version of the bulk states. However, it has been completely disregarded in the present manuscript. The authors prefer to make claims like “we do not exclude bulk bands overlapping with F”. The bulk states should be however well visible –and they are indeed- and they should not overlap with surface states. They are well visible for all other hexaborides (EuB6, CeB6, LaB6, etc.). In the same framework, the authors have ignored my proposition to verify whether a directional projection of the bulk ellipsoids would give the contours in question. A question about the directional projection came from Reviewer 3 as well. The difference in the contours between SmB6 and CeB6 could be perhaps simply explained if the bulk contours of the first are projected on the (111) plane, due to surface confinement, while the contours of the latter are simply cut along the (111) plane. I understand that the projection scenario would be clearly against any topological nature of the states in question but it should have been considered, especially when a new surface orientation is successfully measured.

Finally, I cannot understand the spin polarization observed in Fig. 4(a), after seeing the sketch in Fig. S7(d): it seems to be that there should not be any measurable spin polarization when we cut along $[-1-12]$ because it is the “turning point” from blue to red, hence zero.

All in all, I see that the authors consider no alternatives in the interpretation of their results on equal footing with the topological scenario, I disagree with their claim that these data are a conclusive proof for topological origin and I stay in my position against publication.

Reviewer #3 (Remarks to the Author):

This revised manuscript is greatly improved. In particular I am happy that the following specific concerns below have been addressed:

(i) The authors now cite the various egregious missing references brought to their attention including (a) a previous theory prediction for the (111) surface states, Baruselli Ref.31 regarding the sign of the winding number, (b) a previous spin-ARPES of SmB6 of Suga Ref.39 who observed strong 4f spin-asymmetry induced by incident circular polarized x-rays, and (c) the surface Kondo breakdown scenario, Alexandrov Ref. 4.

(ii) The literature discussion of the dHvA controversy is corrected and citations updated, and the previous claims about providing insight into the 2D vs 3D controversy are now removed (for logical quagmire reasons). Also a discussion and quantification of the possible multiplicity of 2D orbit sizes from the Fermi contour overlapping is now provided in the supplement for completeness.

(iii) The quantitative size comparison of the in-gap Fermi contours to EF-extrapolation of the S2 band and Fig. S5 visualization provides a nice reference for comparison to the (001) surface

results which exhibit much larger k-shifts and relatively smaller in-gap state Fermi contours. Explanation for such differences between SmB6 surface and the curiosity of the larger mixed valent contours than for trivalent CeB6 is beyond the scope of this work, but at least now the issue is acknowledged and out there for someone else to take interest in.

(iv) The Fig. S7 schematic answering the Referee #1 question about the out-of-plane spin-polarization (non-)cancellation in the overlap regions is a nice addition and is consistent with the (111) surface spin texture presented by Baruselli Ref. 31.

- - -

Concerning the figure and discussion of the reinterpretation of the SmB6(001) ARPES data, e.g. by Hlawenka Ref. 14, I see that it is now moved to the supplemental section, and with a softening sentence of "It should be noted that this interpretation is just a possibility based on known results." in response to other referee comments.

I agree with the logic that an umklapp scattering process would NOT predict the experimentally observed different Fermi velocities between the original and umklapp replica bands. In my previous review, I was merely try to add that umklapp scattering would also NOT predict a circular Fermi contour replica if the original contour was elliptical.

Since this discussion was moved to the supplement with no key point remaining in the main text, and with a new softening sentence in the supplement, it is now not appropriate to have in the Summary the strong claim that "Based on these results, a consistent interpretation of the previous, controversial studies performed on the four-fold (001) surfaces has been achieved." I recommend that the authors at least state in the main text the basic summary of what is found in the supplement, e.g. that 'a reinterpretation of an "umklapp" band on the (001) surface allows reconciliation of the odd number count of topological in-gap states, as discussed in more detail in the supplement.'

- - -

Ultimately I am in favor of this manuscript being published based on the strength of the improved spin-ARPES resolution and the novelty of (111) surface removing the ambiguity of the (001) Gamma-bar in-gap state(s). However, even with the above manuscript improvements, there are still two points that prevented me from giving a recommendation for publishing this manuscript version.

(1) The question about the direction of the in-plane polarization being radial (along [-1-12]) or tangential (along [-110]) to the elliptical contour in Fig. S6, seems to be conveniently resolved as labeling mistake. However the original labeling was consistent with azimuthal angle rotation of the sample shown in Fig. 4(b) with an in-plane spin-asymmetry sensitivity along a horizontal x-axis. The corrected labeling now suggests a spin-sensitivity along a vertical y-axis that is inconsistent with Fig. 4(b) and inconsistent with the experimental detection geometry of the HiSOR ESPRESSO machine described in Ref. 38 (Okudo, RSI'16).

In particular, two orthogonal detector target magnetizing coils allow measurement of spin asymmetry along ONE in-plane (x) direction and the out-of-plane (z) direction. According to the figure inset FS schematics, the sample azimuth in Fig. 4(b) and Fig. S6 is rotated relative to Fig. 4(a), but the detector geometry horizontal in-plane spin asymmetry axis remains the same. Thus the spin-asymmetry direction for Fig. 4(b) and Fig. S3 should be the same, i.e. along [-1-12] / G-M as stated in the text for Fig. 4(b).

The small additional polar angle rotation for Fig. S6 would not add a [-110] vertical y-axis component to the spin-asymmetry. The authors need to either need improve their experimental geometry description or provide some rationale for the radial spin-asymmetry in Fig. S6.

(2) The authors newly provide additional photon energy dependent data for SmB6(111) in Fig. S2 from 15-39 eV, as requested by Referee #1. As a result there is a new discussion of how the S2

band outside the Fermi-energy hybridization gap is not a bulk state but instead a surface "resonance" state with citation to a general 1976 review article Ref. 27 that discusses the general definition of surface resonances.

While the authors try to counter the non-topological model for the (001) surface by Hlawenka (Ref. 14) by discussing the ambiguity/controversy of the third/odd-number in-gap state centered on Gamma-bar, the authors miss directly addressing another key part of the Hlawenka model: the observation of 2D character in the states at deeper binding energy outside the in-gap region, and thus a claim that bands S2 and S1 are linked together as part of the same surface state (of non-topological band-bending origin). In other words, I can imagine that group claiming that the similar dimensionality behavior on both (111) and (001) surfaces is further confirmation of their model. The authors need to more explicitly address this issue beyond just the current citation a review article that discusses surface resonances.

The authors have clarified the origin of their strong statement of having "unambiguously solved the problem", with the more explicit statement in the revised manuscript that they are merely following a recipe of determining the non-trivial topological order by counting an odd number of spin non-degenerate surface state Fermi contours. It seems that a corollary to this recipe is that the detailed understanding of and open questions about those odd number single-spin states (e.g. relationship to the S2 'surface resonance') is not important.

Is the logic truly such that there cannot exist an odd number of trivial surface states that exhibit spin-polarization? Is this based on the Rashba spin-splitting effect only generating an even number (pairs) of single-spin bands? The authors seemed to be able to readily rationalize spin polarization of the S2 surface resonance bands via the rebuttal statement that "spin-polarization of S2 is not surprising as a surface band". No similar sentence or citation is however found in the main text. Are the authors also referring to the Rashba effect in this statement?

Finally, I still have a strong negative gut reaction to summary sentence "These results conclude the discussion on the topological order of SmB6" – which in general is a very unscientific phrasing. My understanding is that the "topological order of SmB6" is hardly controversial in the minds of theorists based on the basic d-f band inversion that is experimentally verified long ago. What is controversial is whether the experimental ARPES in-gap states (with no clear Dirac point) and also low T transport properties (of oxidized surfaces) originate from this topological order or whether other non-topological effects are also at play. This manuscript provides a 'strong' case for a direct relation of the (111) in-gap states to the topological physics, and presents (now in the supplement) an alternate interpretation of (001) surface results that is also consistent with the topological origin. However I predict that with the myriad of still open questions, that the scientific literature discussion of the relation of the ARPES in-gap states to the topological nature is far from "concluded".

Reviewer #1 (Remarks to the Author):

The authors have answered, to some extent, my questions and have improved the manuscript. The characterization of the (111) surface state is now more exhaustive, the bands are now clearly labelled and better quantified through MDC-EDC analysis; also, the out of plane spin polarization is now clear and the readability overall has improved, even if it is still not perfect.

However, there are still important shortcomings in this version of the manuscript and I cannot recommend publication in Nature Communications at this stage. My main concerns are related to the analysis/interpretation of the data presented and the strong conclusions that are drawn from it.

My biggest concerns are the following:

1 - Bulk vs surface band structure:

The definition of S2 and F as surface states is still not convincing to me. In the SM the authors now show the photon energy dependence of two MDCs, crossing S1 and crossing S2 in fig. S2(b) and (c) respectively.

The photon energy range spanned in this experiment covers 80% of the Gamma-R distance as discussed by the authors. However the initial photon energy of 15 eV does likely not correspond to the actual bulk Gamma point, and the range is more similar to 50-60% of Gamma-R and 30-20% of R-Gamma (which carries the same information of Gamma-R because of inversion symmetry). More accurately the energy range used spans 40% of the bulk Brillouin zone.

At the same time, while the MDCs for S1, I agree, are more likely to indicate a two dimensional feature, the same cannot be said for S2, where there is a visible shift of the peaks in k_y for curves at, for example, 15,19,20,21,32 eV with respect to the peaks well centred on the green mark at different photon energy. Also, in the curves at 22-24 eV the peaks are not visible, but it looks like they could just be at the edge of the shorter k -range shown. This is likely just some strong background non-uniformity and the peak could not be visible because of the cross section, but the data quality here presented in this k_z range studied is insufficient to conclude that S2 is a surface state.

First of all, let us express our deepest acknowledgement to the reviewer #1. Thanks to his/her comment, we could improve our manuscript (MS) significantly. In the following part, we made point-to-point responses to each comment.

We agree with the reviewer #1 (as well as the other reviewers) that the current data is not enough to show the perfect two-dimensionality of S2. According to the comment, we modified the assignment of S2 and F bands. In the revised version, both bulk and surface origins are proposed and discussed as parallel, equally possible origins (p. 5-6 in the main text and p. 3-4 in SM).

As for this modification, we'd like to emphasize two points. Firstly, as the reviewer #1 stated, the bulk origin of S2 plays no role to determine the topological order, because it is determined solely by the surface states crossing the Fermi level. Secondly, the spin polarizations from S2 or F observed by spin-resolved EDCs (Fig. 4 (a)) is not strange, even if they are bulk bands, because spin-dependent reflectivity (ref. [30] in the revised main text) can provide spin-polarized photoelectrons from bulk bands.

An additional question arises, if these (S2, F, S1) are all surface states why the bulk Sm d- and f- states are not visible in the ARPES spectrum? These are usually very visible in SmB6 at these photon energies. S2 also seems to continue at deeper binding energy, below 0.2 eV (Fig.S2(a)), is this also a surface resonance?

We agree with the reviewer that the Sm-5d bulk states should be observed in our experimental condition. And actually, they are observed as broad humps away from the S2 peaks. In the revised SM, we added triangle markers to indicate them. It must be from Sm bulk bands and more or less agrees with where the reviewer pointed out.

In summary: While it is proven that, on the (001) surface, there can be surface resonances, the arguments for lack of 3D dispersion in S2 is weak in this case because of the limited k_z range spanned, and evidence of MDC peaks centered at different k_y for different photon energies. Furthermore, the absence of 3D bulk bands and the similarity of S2 and F with the respective Sm d- and f- states is suspicious.

For a reader looking at this data, as presented here, a more likely scenario would be that these are the standard d and f bulk states that one would find projecting onto the (111) surface the bulk bands. Be noted that this scenario would of course not affect the possible presence of topological states at the surface.

As explained above, we modified the assignments for S2 and F in the revised version and now present two alternative possibilities, bulk and/or surface origin of them. Note that even if S2 is assigned to be the surface resonance, the bulk Sm-5d bands are observed as the humps away from them, as indicated in the revised Fig. S2. The case is similar for f-

bands. Even if some part of Sm-4f peak is from surface, there must be the bulk-derived intensities at the same time.

2 - Spin-ARPES Analysis:

I am still puzzled by the spin-ARPES data and their interpretation:

It must be noted that acquiring spin-ARPES data within the small gap of SmB6 is a very challenging task, so the data quality (signal to noise ratio) here presented, is notable. However, the spin profiles, here derived from the raw data as described in the methods, are particularly odd:

Fig 4 (a-b) shows the spin-MDC at the Fermi level. I find the shape and position of the peaks, related to the putative topological state, a matter of concern. In panel (a) the left peak (negative k) only shows spin up (nominally, red) component, yielding a $\sim 100\%$ spin polarization; the right peak has both spin up and down components. This is not necessarily concerning because states at opposite momenta could have quantitatively different spin polarization (i.e. [Zhu et al, PRL 112,076802 (2014)]), but here the blue and red peaks have a different shape, in particular the full width half maximum is different as well as the peak location.

At first, we have to correct our data presented in the last MS. The spin-resolved MDC along Γ -M (Fig. 4 (a)) in the last version was obtained after the subtraction of polynomial backgrounds. This should have been stated explicitly, but was missing. We apologize a possible confusion to the reviewers because of this missed explanation.

For the information to the reviewers, here we show the spin-resolved MDC along Γ -M WITH the background as Fig. R1. The monotonously increasing BG would be due to the metal baseplate (Mo) or the other surroundings. Such non-spin-polarized BG was observed only in this SARPES-MDC set. For the other conventional ARPES nor SARPES data, MDCs along Γ -K and EDCs, we did not perform any BG subtractions. The broad hump the reviewer thought to be another peak at $+0.24 \text{ \AA}^{-1}$ would come from the background subtraction for the SARPES MDCs along Γ -M.

Fig. R1

The SARPES MDC along Γ -M with the background.

It would be possible to analyze the peak positions even with such BG, by for example fitting the spectra with BG. However, the information about the spin polarization of S1 along Γ -M could be obtained from SARPES EDCs shown in Fig. 4(a) without any fitting nor BG subtraction. Therefore, we decided to remove the SARPES MDC data along Γ -M from the manuscript and to use the EDCs instead of it, so that the readers could avoid such complex explanation for the quantitative analysis.

This is even more evident in Fig 4.(b): here the blue peak at negative k seem to have different peak location respect to the red peak (shifted to higher k). In the positive k side this is also very obvious, the blue profile shows a peak around 0.35 \AA^{-1} , but it is impossible to define an actual peak position for the red curve. The k range shown is too limited, and an actual peak shape cannot be distinguished. The spin up and down distribution curves should show the same peak shape if they are referred to the same state, because the additional spin selectivity of the technique does not change the spectral function of the state, but merely the intensity. the apparent shift in peak location between opposite spin signals is more indicative of a two components feature (one spin up and one spin down). This scenario is also corroborated by the crossing between red and blue curves at the right edge of Fig. 4(b), but the k range of fig.4(b) is too small and should be extended to provide any meaningful information on the right peak.

Along Γ -K shown in Fig. 4 (b), as the reviewer correctly pointed out, the peak shape is not symmetric. It is because the TSS crosses the Fermi level two times (~ 0.3 and $\sim 0.7 \text{ \AA}^{-1}$). Moreover, the tail from the TSS lying very close to the Γ -K line is also observed as a background. To make the situation clearer, we provide some comparison data from conventional ARPES in Figs. S8 (a), (b). From them, one can see three components, two peaks corresponding to k_F and the broad background from the neighboring TSS.

The observed MDC along Γ -K by SARPES generally agrees with the conventional ARPES

data. Slight difference of the peak positions and shapes would be due to the wider energy and angular resolution of SARPES. From this comparison, one can see the spectral shape of SARPES MDC along Γ -K is not due to any inaccuracy of the SARPES measurements but from the intrinsic surface electronic structure. From the in-plane and out-of-plane spin polarizations for the SARPES MDCs, one can also see that the TSS at k_F ($\sim 0.3 \text{ \AA}^{-1}$) is spin polarized. The detailed discussion is shown in the revised SM (p. 9-11).

We agree with the reviewer that to obtain the complete, quantitative information of the photoelectron spin polarization is difficult from the current data. However, we do not think it is the crucial information to determine the topological order of SmB_6 , because one only needs to exclude the doubly spin-degenerate states to determine the topological order from the metallic surface states. Such complete spin-polarization measurement would be an interesting, but different future work apart from the current manuscript.

Secondarily, the authors have stated in the rebuttal that the spin-ARPES signal does not undergo any renormalization. Does that imply that raw data curves are not renormalized with respect to detector efficiency (namely, in the VLEED, the different scattering efficiency of the target for up and down magnetization)? Does that mean the detector is perfectly equally efficient for all opposite spins? This question arises looking at the EDCs. It is obvious that there is some polarization somewhere but, as stated by the authors, all bands in these plots seem to be spin polarized. This is surprising, also in view of the discussion about bulk vs surface states. Generally, the spin-ARPES normalization process can be quite complicated and it benefits from the presence of spin degenerate states that show no polarization or a clear opposite spin band as comparison.

I encourage the authors to present the spin-ARPES data more clearly with, if needed, ulterior analysis. The k-energy range should be large enough to fully distinguish the peaks considered and, if the opposite spin signals show non-equivalent peak shapes, this should be justified. A spin-degenerate background or state would also make the analysis more convincing.

Yes. The efficiency of the VLEED detector is almost completely insensitive to the magnetization of the target, for spin-degenerate electrons.

The SARPES EDCs of cut 1-2 and 5 (Fig. 4 (a)) has the opposite sign of the wavevector and the sign of the spin polarization is inverted. This is exactly what the reviewer requested, “a clear opposite spin band as comparison”.

Moreover, for the information to the reviewer, we show the SARPES EDCs obtained by the same VLEED detectors as Fig. R2. Here, the spectra at the center of the surface

Brillouin zone (0 deg.) show no polarization. At ± 2 deg., a clear opposite spin bands are also observed as comparison. All of these SARPES spectra were obtained without any renormalization process.

For the reviewer's information, the renormalization for SARPES is mostly required for the Mott-type spin detector, because it uses a pair of detectors for each spectrum. In contrast, the VLEED-type detector uses exactly the same electron detector and electron-diffraction geometry. The only difference is the magnetization of the target. Therefore, the VLEED-type detector intrinsically requires less (or even no, as in our case) renormalization to cancel out the artificial asymmetry between each spin channel.

Fig. R2

The SARPES EDCs of the Bi(111) ultrathin films observed by the same SARPES setup as this work. The data is from PRB97, 085433 (2018).

As for the data processing, we modified the explanation in p. 10 (methods) to be more precise. The raw set of SARPES spectra is composed by the four spectra, (I_p , I_n , I_n , I_p), measured by this order. It is in order to compensate the time-dependent degradation of the surface states as well as the decay of incident photon flux (the beam current of HiSOR starts decaying slowly after injection). This procedure is the same for the data in Fig. R2. This process is not the renormalization as required for the SARPES analysis from MOTT detector, because we don't need any arbitral coefficient.

3 - Conclusions

I renew my scepticism when the authors state that their work “conclude the discussion on the topological order in SmB₆”. This is not simply because of the band assignment and spin-ARPES data as previously discussed.

Even if these points are clarified, Spin-/ARPES is usually insufficient to conclusively define a topological insulator phase. ARPES may lack the ability of detecting some electronic states for several reasons (resolution, range, matrix elements). For this reason, while ARPES is a great tool to provide evidences of a topological insulator phase, and has been successfully employed in many cases, it will likely not provide a definitive answer, if further techniques are not employed. This is particularly difficult for a complicated system like SmB₆, where the correlation gap is very small, and the samples are well known to be unreliable. A great number of contrasting results have been shown throughout the years with evidence both in favor and against the topological picture [Xu et al. *J. Phys.: Condens. Matter* 28,363001(2016)], and this study, while interesting and important, could not fundamentally provide a definite conclusion to the SmB₆ case without a systematic study of different samples and with different techniques.

As the reviewer commented that some phrases such as “conclude” might appear too assertive. Therefore, we modified our statements in the revised MS, especially in the abstract and summary.

The reviewer commented that “Spin-/ARPES is usually insufficient to conclusively define a topological insulator phase. ARPES may lack the ability of detecting some electronic states for several reasons (resolution, range, matrix elements).” However, we obtained results presented in this manuscript using some of the highest quality Spin-/ARPES setups and also used the highest quality SmB₆ sample without any other metallic surface states.

The concerns posed by the reviewer #1 are excluded as follows,

- Resolution

The energy resolution of the conventional ARPES is 15 meV, smaller than the bulk Kondo gap. This value is enough to observe the in-gap states, as done in this work.

- Range

The photon-energy range in this work is enough to cover the region where the photoexcitation cross section for Sm 5d, 4f, and B 2sp bands are large enough to observe the photoelectrons from them. Therefore, any states from SmB₆ should be observed in this condition, if exists.

- Matrix element

For the conventional-ARPES measurement, we used off-normal incident circularly polarized photons and summed up both right- and left-handed spectra. By this, all the circular dichroism is cancelled out. Moreover, the photon-incident plane is slightly away from the high-symmetry plane (this explanation is added in p. 2 of SM). Therefore, no symmetry operation in the photoexcitation matrix element can vanish the photoelectrons from surface states. For the final-state effect, it could vanish the states at an incident photon energy, but not for a range of energies as shown in Fig. S2.

Our sample is prepared by the same method as ref. [39], whose bulk electronic properties were extensively tested. We found no clear point to doubt the reliability of our sample. If not, the reviewer should provide the possible problems from the sample explicitly.

As explained above, we have performed the reasonable range of experiments and consideration to exclude the other surface states. Based on the obtained results without any prejudice, we are quite sure that the readers could accept our conclusion of the non-trivial topological order of SmB₆. Of course, our statement could be contradicted by the new results. However, at present, we believe many readers can believe our conclusion.

As I have previously stated I think that the study of the (111) surface of SmB₆ is very interesting, important and timely and can contribute to the heated discussion on the topological nature of the material. While the data could be worth publication, the present manuscript still has some critical shortcomings.

We would like to thank the reviewer for the high evaluation. According to the comments by the reviewer #1 as well as the other reviewers, we amended the manuscript. We hope all the shortcomings from the manuscript are now removed and the current version is suitable for publication.

Reviewer #2 (Remarks to the Author):

I have read the revised manuscript by Ohtsubo et al., all referee's comments and the authors' rebuttal. The new manuscript has no fundamental changes with respect to its previous version and the authors stay very strongly in their position that their findings are a conclusive proof for the topological nature of SmB₆.

I disagree. I find the association of all bands to surface states very hasty. Reviewer 1 and myself expressed our concerns about that. The authors stay in their position that the lack of k_z dispersion for S1 and S2 is the underlying proof for their surface origin even if –at least S2– has been attributed to the bulk 5d states of Sm in many previous studies and has counterparts in all other hexaborides. As for attributing S1' (or F) to a surface band because of the observation of spin polarized peaks, this is not a convincing argument knowing that the Sm 4f bulk states should be exactly at the corresponding E-k position. Lack of k_z dispersion has been observed for states that are well known not to be new surface states. This can be due to the fundamental limitation of ARPES in resolving momentum components along the surface normal (see Stroscov 2003), or because the states in question are a confined version of bulk bands (e.g. quantum well states). This is exactly what happens in 2DEGs on the surface of perovskite oxides. Such a scenario has been considered by Hlawenka et al. when they discussed a “surface-localized” version of the bulk states. However, it has been completely disregarded in the present manuscript. The authors prefer to make claims like “we do not exclude bulk bands overlapping with F”. The bulk states should be however well visible –and they are indeed– and they should not overlap with surface states. They are well visible for all other hexaborides (EuB₆, CeB₆, LaB₆, etc.).

First of all, let us express our deepest acknowledgement to the reviewer #2. Thanks to his/her comment, we could improve our manuscript (MS) significantly. In the following part, we made point-to-point responses to each comment.

We agree with the reviewer #2 (as well as the other reviewers) that the current data is not enough to show the perfect two-dimensionality of S2 and F. According to the comment, we modified the assignment of these bands. In the revised version, both bulk and surface origins are proposed and discussed as parallel, equally possible origins (p. 5-6 in the main text and p. 3-4 in SM).

As for this modification, we'd like to emphasize two points. Firstly, as the reviewer #1

stated, the bulk origin of S2 plays no role to determine the topological order, because it is determined solely by the surface states crossing the Fermi level. Secondly, the spin polarizations from S2 or F observed by spin-resolved EDCs (Fig. 4 (a)) is not strange, even if they are bulk bands, because spin-dependent reflectivity (ref. [30] in the revised main text) can provide spin-polarized photoelectrons from bulk bands.

In the same framework, the authors have ignored my proposition to verify whether a directional projection of the bulk ellipsoids would give the contours in question. A question about the directional projection came from Reviewer 3 as well. The difference in the contours between SmB6 and CeB6 could be perhaps simply explained if the bulk contours of the first are projected on the (111) plane, due to surface confinement, while the contours of the latter are simply cut along the (111) plane. I understand that the projection scenario would be clearly against any topological nature of the states in question but it should have been considered, especially when a new surface orientation is successfully measured.

On this point, we do not agree with the reviewer #2.

We have never ignored his/her comment but made a clear reply as follows in the last rebuttal letter as follows: (rebuttal letter: p. 11)

The similar appearance of the M -bar state on (111) and the X -bar on (001) would suggest the common origin of them. But it does not play any role to consider the topological order, as discussed above. We agree with the reviewer that the microscopic origin of them might also be an interesting subject, but we think this subject should be discussed elsewhere to focus on the main theme of this MS, the topological order of SmB6.

The possible origin of the surface state from projection and surface confinement is NOT the confliction against the topological order. The microscopic origin of the surface states is irrelevant to the topological classification. In other words, if the odd number of spin-polarized FCs, derived from surface confinement for example, appeared around surface TRIMs, the non-trivial topological order of the bulk electronic structure is proved by them without any conflict.

For example, the partial charges of the interface state of HgTe has a long tail to the deeper layers (e.g. Fig. 3 of PRL105, 176805). Such character is quite similar to the confined 2DEG at the LAO/STO interface (Fig. 2 of PRB 79, 245411). Both of them also shows quite similar dispersions and orbital characters to their “mother” bulk states. However, at the same time, the 2D state of HgTe is the typical topological surface state

reflecting the non-trivial topological order of HgTe.

To avoid the similar misunderstanding by the readers, we added a discussion in p. 9.

Finally, I cannot understand the spin polarization observed in Fig. 4(a), after seeing the sketch in Fig. S7(d): it seems to be that there should not be any measurable spin polarization when we cut along $[-1-12]$ because it is the “turning point” from blue to red, hence zero.

Figure S7 in SM is about the out-of-plane spin polarization. In Fig. 4 (a), the in-plane spins were detected. This point had been already written in the main text of SM. In the current version, we added the same information in the caption of Fig. S7 (p. 9).

For the information for the reviewer #2, we also made the figure R3 to show the in-plane spins with symmetry operations. As shown there, the in-plane spins are not cancelled out along $[-1-12]$.

Fig. R3 Schematic drawings of the spin-polarized FCs with symmetry operations. Here, the in-plane spin polarizations are considered. (a) Single FC at a M point. (b) FCs multiplied by three-fold rotation. (c) FCs multiplied by time inversion. (d) FCs multiplied by the combination between (b) and (c).

All in all, I see that the authors consider no alternatives in the interpretation of their results on equal footing with the topological scenario, I disagree with their claim that these data are a conclusive proof for topological origin and I stay in my position against publication.

As for the topological order of SmB_6 , we have never changed our conclusion, simply because no reasonable shortcomings on this point have been pointed out by all the reviewers.

For the reviewer's information, we have modified the rhetoric in abstract and summary in order to change some too assertive phrases. It is according to the comments from reviewers #1 and #3.

Reviewer #3 (Remarks to the Author):

This revised manuscript is greatly improved. In particular I am happy that the following specific concerns below have been addressed:

(i) The authors now cite the various egregious missing references brought to their attention including (a) a previous theory prediction for the (111) surface states, Baruselli Ref.31 regarding the sign of the winding number, (b) a previous spin-ARPES of SmB6 of Suga Ref.39 who observed strong 4f spin-asymmetry induced by incident circular polarized x-rays, and (c) the surface Kondo breakdown scenario, Alexandrov Ref. 4.

(ii) The literature discussion of the dHvA controversy is corrected and citations updated, and the previous claims about providing insight into the 2D vs 3D controversy are now removed (for logical quagmire reasons). Also a discussion and quantification of the possible multiplicity of 2D orbit sizes from the Fermi contour overlapping is now provided in the supplement for completeness.

(iii) The quantitative size comparison of the in-gap Fermi contours to EF-extrapolation of the S2 band and Fig. S5 visualization provides a nice reference for comparison to the (001) surface results which exhibit much larger k-shifts and relatively smaller in-gap state Fermi contours. Explanation for such differences between SmB6 surface and the curiosity of the larger mixed valent contours than for trivalent CeB6 is beyond the scope of this work, but at least now the issue is acknowledged and out there for someone else to take interest in.

(iv) The Fig. S7 schematic answering the Referee #1 question about the out-of-plane spin-polarization (non-)cancellation in the overlap regions is a nice addition and is consistent with the (111) surface spin texture presented by Baruselli Ref. 31.

First of all, let us express our deepest acknowledgement to the reviewer #3. Especially his/her thoughtful remarks about the improvements in the revised MS encourages us a lot. In the current version, we could improve our manuscript (MS) significantly again, thanks to his/her comment. In the following part, we made point-to-point responses to each comment.

Concerning the figure and discussion of the reinterpretation of the SmB6(001) ARPES data, e.g. by Hlawenka Ref. 14, I see that it is now moved to the supplemental section, and with a softening sentence of “It should be noted that this interpretation is just a possibility based on known results.” in response to other referee comments.

I agree with the logic that an umklapp scattering process would NOT predict the experimentally observed different Fermi velocities between the original and umklapp replica bands. In my previous review, I was merely try to add that umklapp scattering would also NOT predict a circular Fermi contour replica if the original contour was elliptical.

Since this discussion was moved to the supplement with no key point remaining in the main text, and with a new softening sentence in the supplement, it is now not appropriate to have in the Summary the strong claim that “Based on these results, a consistent interpretation of the previous, controversial studies performed on the four-fold (001) surfaces has been achieved.” I recommend that the authors at least state in the main text the basic summary of what is found in the supplement, e.g. that ‘a reinterpretation of an “umklapp” band on the (001) surface allows reconciliation of the odd number count of topological in-gap states, as discussed in more detail in the supplement.’

We agree with the reviewer that such introduction phrase in the main text is important. However, in the last version of MS (p. 8 of the Main text), we have already written the sentences as follows:

At first glance, this conclusion appears to conflict against a recent high-resolution ARPES data on (001) [14]. However, they could be reconciled by an interpretation of the FCs observed on (001). Detailed discussion on this point is shown in SM [26].

In the current version, we only changed the reference number ([26] to [27]). We think the sentences are more or less the same as what the reviewer proposed.

Ultimately I am in favor of this manuscript being published based on the strength of the improved spin-ARPES resolution and the novelty of (111) surface removing the ambiguity of the (001) Gamma-bar in-gap state(s). However, even with the above manuscript improvements, there are still two points that prevented me from giving a recommendation for publishing this manuscript version.

(1) The question about the direction of the in-plane polarization being radial (along [-1-12]) or tangential (along [-110]) to the elliptical contour in Fig. S6, seems to be

conveniently resolved as labeling mistake. However the original labeling was consistent with azimuthal angle rotation of the sample shown in Fig. 4(b) with an in-plane spin-asymmetry sensitivity along a horizontal x-axis. The corrected labeling now suggests a spin-sensitivity along a vertical y-axis that is inconsistent with Fig. 4(b) and inconsistent with the experimental detection geometry of the HiSOR ESPRESSO machine described in Ref. 38 (Okudo, RSI16).

In particular, two orthogonal detector target magnetizing coils allow measurement of spin asymmetry along ONE in-plane (x) direction and the out-of-plane (z) direction. According to the figure inset FS schematics, the sample azimuth in Fig. 4(b) and Fig. S6 is rotated relative to Fig. 4(a), but the detector geometry horizontal in-plane spin asymmetry axis remains the same. Thus the spin-asymmetry direction for Fig. 4(b) and Fig. S3 should be the same, i.e. along $[-1-12]$ / G-M as stated in the text for Fig. 4(b).

The small additional polar angle rotation for Fig. S6 would not add a $[-110]$ vertical y-axis component to the spin-asymmetry. The authors need to either need improve their experimental geometry description or provide some rationale for the radial spin-asymmetry in Fig. S6.

The reviewer is right. With the setup shown in RSI2011 (ref. [41] in the current version), we cannot measure the SARPES spectra shown in this MS. Actually, the ESPRESSO machine had been updated after RSI2011 and now has two VLEED detectors whose electron-incident orientations are 90 deg. rotated with respect to each other, as shown in ref. [43] (Okuda et al., JESRP 2015).

To explain this point, we added ref. [43] and a sentence to explain a pair of VLEED detectors in p. 10 (Methods section).

(2) The authors newly provide additional photon energy dependent data for SmB6(111) in Fig. S2 from 15-39 eV, as requested by Referee #1. As a result there is a new discussion of how the S2 band outside the Fermi-energy hybridization gap is not a bulk state but instead a surface “resonance” state with citation to a general 1976 review article Ref. 27 that discusses the general definition of surface resonances.

While the authors try to counter the non-topological model for the (001) surface by Hlawenka (Ref. 14) by discussing the ambiguity/controversy of the third/odd-number in-gap state centered on Gamma-bar, the authors miss directly addressing another key part of the Hlawenka model: the observation of 2D character in the states at deeper binding energy outside the in-gap region, and thus a claim that bands S2 and S1 are linked together as part of the same surface state (of non-topological band-bending origin). In

other words, I can imagine that group claiming that the similar dimensionality behavior on both (111) and (001) surfaces is further confirmation of their model. The authors need to more explicitly address this issue beyond just the current citation a review article that discusses surface resonances.

According to the comments from the other reviewers, we modified the assignment of S2 and F in the current version. In the revised version, both bulk and surface origins are proposed and discussed as parallel, equally possible origins (p. 5, 7 in the main text and p. 3-4 in SM).

As for this modification, we'd like to emphasize two points. Firstly, as the reviewer #1 stated, the bulk origin of S2 plays no role to determine the topological order, because it is determined solely by the surface states crossing the Fermi level. Secondly, the spin polarizations from S2 or F observed by spin-resolved EDCs (Fig. 4 (a)) is not strange, even if they are bulk bands, because spin-dependent reflectivity (ref. [30] in the revised main text) can provide spin-polarized photoelectrons from bulk bands.

About the second part of the claim by Hlawenka, it has no influence to determine the topological order. The mechanism how the 2D states localized in surface layers formed does not change the topological order. Please also see the reply to the 2nd point of the reviewer #2 (p. 10 in this rebuttal letter).

The authors have clarified the origin of their strong statement of having “unambiguously solved the problem”, with the more explicit statement in the revised manuscript that they are merely following a recipe of determining the non-trivial topological order by counting an odd number of spin non-degenerate surface state Fermi contours. It seems that a corollary to this recipe is that the detailed understanding of and open questions about those odd number single-spin states (e.g. relationship to the S2 ‘surface resonance’) is not important. Is the logic truly such that there cannot exist an odd number of trivial surface states that exhibit spin-polarization? Is this based on the Rashba spin-splitting effect only generating an even number (pairs) of single-spin bands?

Exactly. The detailed understanding of the single-spin state is not important for determining the topological order, as discussed above. And yes, odd number of trivial surface states with spin polarization cannot exist on any insulating materials, according to the general theory of topological insulators. Please see some theoretical reviews (e.g. Ref. [37], Hasan and Kane, RMP82) for details.

The authors seemed to be able to readily rationalize spin polarization of the S2 surface resonance bands via the rebuttal statement that “spin-polarization of S2 is not surprising as a surface band”. No similar sentence or citation is however found in the main text. Are the authors also referring to the Rashba effect in this statement?

In general, surface electronic states are spin polarized. It is sometimes because of the Rashba effect, and sometimes due to the valley polarization, as the cases of transition-metal-dichalcogenides. The common origin of them is the lack of space-inversion symmetry in surface layers.

Even if S2 (and F) were bulk bands, a spin-dependent reflectivity (ref. [30] in the revised main text) can provide spin-polarized photoelectrons from bulk bands.

These discussions were added in p. 6 in the main text.

Finally, I still have a strong negative gut reaction to summary sentence “These results conclude the discussion on the topological order of SmB6” – which in general is a very unscientific phrasing. My understanding is that the “topological order of SmB6” is hardly controversial in the minds of theorists based on the basic d-f band inversion that is experimentally verified long ago. What is controversial is whether the experimental ARPES in-gap states (with no clear Dirac point) and also low T transport properties (of oxidized surfaces) originate from this topological order or whether other non-topological effects are also at play. This manuscript provides a ‘strong’ case for a direct relation of the (111) in-gap states to the topological physics, and presents (now in the supplement) an alternate interpretation of (001) surface results that is also consistent with the topological origin. However I predict that with the myriad of still open questions, that the scientific literature discussion of the relation of the ARPES in-gap states to the topological nature is far from “concluded”.

As the reviewer commented, some phrases such as “conclude” might appear too assertive. According to the advice from the reviewer, we modified our statements in the revised MS, especially in the abstract and summary.

We hope this modification could solve the negative gut reaction of the reviewer #3.

In conclusion, we have provided comprehensive answers to all questions and comments brought by all reviewers. We hope that all reviewers agree with the publication of this manuscript in Nature Communications.

Reviewers' comments:

Reviewer #1 (Remarks to the Author):

The authors have answered extensively my questions.

While the manuscript has again improved overall, the analysis and conclusion, in particular of the spin-ARPES data, still appears not to be rigorously done. This can be partly blamed on the difficulty of interpretation of spin-ARPES data but it appears to me that the authors have extracted excessive conclusions from a somehow confusing and unreliable dataset.

My main concern remains, and it's mainly related to the interpretation of S2, F and S1. While, according to the authors, the main message of the paper only concerns the putative TSS, S1, the data presented on S2 and F are of extreme importance. In fact, these states either represent the bulk bands that give origin to the Kondo gap, or they stem from some sort of surface resonance of the bulk bands, which, as presented in ref 14. can also appear within the gap. Here the authors do not conclude on the origin of these states based on their data, however they do show some possible surface + bulk component of S2 in Fig. S2, suggesting this is a surface resonance. Just as for S1, S2 also is spin polarized and does not have a spin-split pair band, that should exist for any trivial state. However, S1 and S2 in the paper are conveniently regarded in different terms on the basis that S1 is found within the Kondo gap, but this is not fully justifiable if S1 is also a surface resonance. Moreover, the spin-ARPES data also show a strong spin polarized background that renders the data analysis very challenging.

This said, I will not oppose publication in Nat. Comm. because, whether the interpretation is correct or not, the data are new and of broad interest and will spark renewed discussion in the issue of Smb6.

Reviewer #2 (Remarks to the Author):

I have read the revised version of the manuscript by Ohtsubo et al. Here below are my comments:

- I see that the authors have taken one step back regarding the assignment of S2 and F bands to surface features and they now consider both bulk and surface contributions. This is appreciated, but from previous studies it has been well established that at the energy range of the S2 and F features, bulk bands should be rather the dominant spectral features and not weak humps on the EDCs. I therefore agree with reviewer 1 that S2 and F are most probably the standard d and f states projected onto the (111) surface.

- Considering the origin of S1, I wish to further clarify the comment I made on my previous review. Let's consider the possibility that S1 is simply the continuation of S2 with a very small hybridization gap between them. If this is the case, all three bands would satisfy the argument made in the first paragraph, as all of them would have a dispersion expected from bulk bands projected on the (111) surface. Of course the same could be true for data acquired on the (100) surface if one is ready to accept a very small hybridization gap that lies not at the Fermi level but 20 to 30 meV below. In fact, activation energies as low as 3-5meV have been proposed for the hybridization gap (see e.g. Wolgast PRB 2013, Flachbart PRB 2001), but the ARPES community insists that the gap must be much larger since the very first data on the (100) surface. I agree though with Ohtsubo et al. that S1 is surface-confined. However, I disagree that the origin of this surface-confined state is unimportant before assigning to it a topological character. I apologize for not making myself clear before but I am not referring to the bulk extension of the wavefunction. I am referring on the possibility that this surface-confined state defines the upper edge of the hybridization gap rather than being inside the hybridization gap. If the surface-confined state is a confined version of the bulk conduction band (e.g. a quantum well state as in transition metal oxide surfaces and interfaces), then it is this state that defines the upper edge of the gap and

hence it cannot be an in-gap state. If it cannot be an in-gap state, then it cannot be a topological surface state. Hlawenka et al. have considered this possibility and have concluded that this state is indeed an energy-shifted, surface-localized d-f hybrid that it is not confined to the bulk hybridization gap. Ohtsubo et al. do not consider this possibility at all and this is very crucial because such a scenario is incompatible with assigning a topological character to these states. I believe that it would be fairer to consider both scenarios (in-gap topological vs. surface-confined but not in-gap) on an equal footing and the reader could judge by himself. The observation of spin polarization for S1 does not change much of the above argument as spin polarization could be very well due to spin-dependent reflectivity as for S2 and F. Therefore, as it stands now, I cannot recommend the manuscript for publication.

I thank the authors for their clarification on my other comment on the spin polarization.

Reviewer #3 (Remarks to the Author):

This second revision of the manuscript still has some issues to comment on.

(1) The responses to Referee #1 and #2 concerns over the S2 and F labeling and discussion were illuminating to learn what details were missing and not discussed in earlier drafts.

(2) Similarly, it is very disappointing that the authors failed to properly describe their experimental setup and did not reference their current upgraded configuration of two VLEED detectors [Ref. 43] until two rounds into the review process. If they had done so earlier, then a natural question to have been asked is why do the authors not offer a complete presentation of all three spin-components for any particular k-point? Since the VLEED/exchange scattering is a serial measurement of each individual spin-component (with a 4-step serial measurement of opposite target magnetizations as now described), did the authors specifically only measured a subset of the 3 vector components or did the authors measure all three and are only presenting the subset relevant to their spin texture model?

I am willing to now accept the earlier "mislabeling" answer to the previous initial draft issue of the spin-asymmetry direction in Fig. S1 spectrum. It would be nice if the authors also added the 2015 double VLEED reference [43] to their supplemental description of the experimental geometry, and in the description of Fig. S6. provided the information (if correct) that in addition to rotating the azimuth angle of the sample relative to Fig. 4(a,c,d), the "other" VLEED detector was used to obtain the "vertical" in-plane component, i.e. orthogonal to the "horizontal" component measured in Fig. 4(a,c,d). Fig. 4(b) appears to be measured with the sample azimuth rotated relative to Fig. 4(a,c,d) panels and with the same VLEED detector for the "horizontal" in-plane component.

(3) The authors did not fully understand my technical complaint concerning the summary sentence about "a consistent interpretation . . . has been achieved." There is a general guideline in many journals (although I do not see it explicitly mentioned for this journal) that the main manuscript should stand on its own and be understandable without the supplement. There is no problem with stating that some other interpretation of the (001) surface ARPES is described in the supplement with no further details given in the main manuscript, e.g. merely "...reconciled by an interpretation...". However if one is going to then use the results of that supplementary discussion in the conclusion summary paragraph as a key point with the very assertive "consistent interpretation . . . has been achieved", then this sentence violates the stand-on-its-own guideline. I.e. the reader must go to the supplement to understand what this "achievement" is all about, since the sentences a few paragraphs earlier calling attention to the topic provides no specific details beyond "an interpretation". What I am asking is that the authors be consistent with all their other references to the supplement ("SM") in which it is stated clearly what is to be found there. I offered an example recommendation of such specific details in the previous review. Also in the similar vein of complaints about "being too assertive", the word choice "has been achieved" should be changed "has been proposed".

(4) The authors newly use the term "many-body resonance" as a shortcut to describe origin of the (001) in-gap states in the Hlawenka model [Ref. 14]. Those authors actually used the term "many-body resonance...shift", i.e. the "many body resonance" refers to the 4f peak and the "shift" refers to the energetic response (in the first sub-surface layer) to the various termination-dependent surface charges. Hence it falls into the category of "surface polarity", but distinctly different from Zhu [Ref. 36] whose model is more a combination of "dangling-band + surface polarity". The authors should use a different descriptive term for the Hlawenka model. It is noteworthy that the relaxed (110) and (111) surfaces are not necessarily immune to similar near-surface polarity-induced energy shifts.

(5) The authors now explicitly state "that the detailed origin of the surface states, such as ..., plays no role for the (topological) classification". While I understand where this statement is coming from, my opinion is that in the real world of scientific argumentation, it does matter. If in the future some new compelling evidence is presented that the (111) surface in-gaps states have a surface polarity origin, AND that origin does not have a clear theoretical basis for generating single-spin states, or has no connection to parity inversion of bulk bands, then one has a conflict that needs to be resolved. One either has to refine the origin model or maybe look again at the link between single-spin states and the origin of the measured partial spin-polarization, such as the possibility described in the newly added reference [30] of different spin-dependent reflectivities of Bloch waves off the surface that generates spin-polarization in otherwise spin-degenerate bands. In that sense, the type of surface state origin does matter.

I can continue to recommend this manuscript for publication for its contribution to the on-going debate on SmB₆, even if my confidence in the result tends to weaken after each review stage.

First of all, let us express our deepest acknowledgement again to all the reviewers and the editor. Thanks to their efforts, we could improve our manuscript (MS) again. As a major update in this version, we have added a paragraph (p. 9) to discuss the limit and possible update of our topological classification in this work. We believe that the current version is acceptable for general readers.

In the following part, we made point-to-point responses to each comment.

Reviewer #1 (Remarks to the Author):

We'd like to thank the reviewer again for the fruitful discussion to improve our MS. We are happy that finally he/she encouraged the publication of our MS, while pointing out some concerns. Here, we'd like to provide our answers to the reviewer's concerns. These comments/replies would be published together with the accepted MS itself in Nature Communications. So, we hope this communication would also be helpful for the readers and the following researches in the issue of SmB₆.

(1)

The authors have answered extensively my questions. While the manuscript has again improved overall, the analysis and conclusion, in particular of the spin-ARPES data, still appears not to be rigorously done. This can be partly blamed on the difficulty of interpretation of spin-ARPES data but it appears to me that the authors have extracted excessive conclusions from a somehow confusing and unreliable dataset. My main concern remains, and it's mainly related to the interpretation of S₂, F and S₁. While, according to the authors, the main message of the paper only concerns the putative TSS, S₁, the data presented on S₂ and F are of extreme importance. In fact, these states either represent the bulk bands that give origin to the Kondo gap, or they stem from some sort of surface resonance of the bulk bands, which, as presented in ref 14. can also appear within the gap. Here the authors do not conclude on the origin of these states based on their data, however they do show some possible surface + bulk component of S₂ in Fig. S2, suggesting this is a surface resonance.

We'd like to emphasize again that the topological classification of the material (ref. 22, PRB78, 045426, ref. 36, RMP82, 3045), SmB₆ in this case, to be a topological or normal insulator does never depend on the character of its surface state(s). It is the same for the surface "resonance" case the reviewer pointed out here. The "resonance" means a metallic electronic state localized in the surface atomic layers. It is nothing but a metallic surface state, according to the simple topological classification procedure.

However, in order to meet the repeated concerns by all the reviewers and the editor, we added a paragraph in the discussion part (p. 9) so that the reader could understand the principle we are standing on and its possible limitation as pasted below.

The topological classification procedure assumes weakly correlated insulator. Therefore, we cannot exclude a possible violation of such simple topological classification by strong electron correlation. To the best of our knowledge, such work has never been published so far. However, once such discovery has been achieved, the topological classification performed in this work should be revisited.

Moreover, according to the requests from the reviewers and the editor, we added a section in the Supplementary Material (SM) to discuss the possible origin of the surface states observed in this work (p. 13-14 in SM). The surface "resonance" case, it would be based on the "many-body resonance" proposed by Hlawenka et al., is also discussed there.

(2)

Just as for S1, S2 also is spin polarized and does not have a spin-split pair band, that should exist for any trivial state. However, S1 and S2 in the paper are conveniently regarded in different terms on the basis that S1 is found within the Kondo gap, but this is not fully justifiable if S1 is also a surface resonance.

In the projected bulk bands, the electronic state localized in the surface layers can be observed with spin polarization without its partner. It is because the partner could be merged into the bulk bands. At least, the degree of hybridization with bulk bands could be different in each spin-split branches of the state, justifying the lonely spin-polarized branch in the projected bulk bands. An example was found in the subsurface layers of Ge (PRB88, 245310).

(3)

Moreover, the spin-ARPES data also show a strong spin polarized background that renders the data analysis very challenging.

We agree with the reviewer that the analysis of the spin-resolved ARPES data and its interpretation is sometimes very difficult. For the discussion following to the publication of this manuscript (hopefully), we'd like to point out that,

1. we've provided detailed experimental geometries as well as the procedure how we analyze the raw SARPES spectra so that the following researchers could compare the new results with ours, and
2. one just needs to verify that the observed surface state is not doubly spin degenerated, for the topological classification we're standing on in this work.

(4)

This said, I will not oppose publication in Nat. Comm. because, weather the interpretation is correct or not, the data are new and of broad interest and will spark renewed discussion in the issue of SmB₆.

Together with the reviewer, we hope our work could provide a new and important information for future discussion and progress in the issue of topological Kondo insulator and the peculiar electronic phenomena of SmB₆.

Reviewer #2 (Remarks to the Author):

We'd like to thank the reviewer again for the fruitful discussion to improve our MS.

I have read the revised version of the manuscript by Ohtsubo et al. Here below are my comments:

(1)

- I see that the authors have taken one step back regarding the assignment of S2 and F bands to surface features and they now consider both bulk and surface contributions. This is appreciated, but from previous studies it has been well established that at the energy range of the S2 and F features, bulk bands should be rather the dominant spectral features and not weak humps on the EDCs. I therefore agree with reviewer 1 that S2 and F are most probably the standard d and f states projected onto the (111) surface.

As pasted above, the reviewer #1 is now assuming the S2 and F to be surface "resonance" one, which has a similar origin to the bulk bands, which would be $5d$ and $4f$, respectively, and slightly perturbed in the surface atomic layers. We think such different assignment between the reviewers is the clear evidence that it is very difficult to assign the origin of S2 and F bands from the current dataset. Therefore, we believe that the current presentation, to show two different possible assignments of S2 and F (bulk and surface) and the corresponding interpretations of the SARPEs data, is the fairest way to introduce the observed information of S2 and F.

In addition, we'd like to point out that we do NOT write that the broad humps on the MDCs in Fig. S2, derived from bulk bands, are weak. For example, at 32 eV, the broad hump at $\sim +0.35$ ($1/\text{\AA}$) has the larger peak area than the rather sharp S2 peak at ~ 0.2 ($1/\text{\AA}$).

(2)

- Considering the origin of S1, I wish to further clarify the comment I made on my previous review. Let's consider the possibility that S1 is simply the continuation of S2 with a very small hybridization gap between them. If this is the case, all three bands would satisfy the argument made in the first paragraph, as all of them would have a dispersion expected from bulk bands projected on the (111) surface. Of course the same could be true for data acquired on the (100) surface if one is ready to accept a very small hybridization gap that lies not at the Fermi level but 20 to 30 meV below. In fact, activation energies as low as 3-5meV have been proposed for the hybridization gap (see e.g. Wolgast PRB 2013, Flachbart PRB 2001), but the ARPES community insists that the gap must be much larger since the very first data on the (100) surface. I agree though with Ohtsubo et al. that S1 is surface-confined. However, I disagree that the origin of this surface-confined state is unimportant before assigning to it a topological character. I apologize for not making myself clear before but I am not referring to the bulk extension of the wavefunction. I am referring on the possibility that this surface-confined state defines the upper edge of the hybridization gap rather than being inside the hybridization gap. If the surface-confined state is a confined version of the bulk conduction band (e.g. a quantum well state as in transition metal oxide surfaces and interfaces), then it is this state that defines the upper edge of the gap and hence it cannot be an in-gap state. If it cannot be an in-gap state, then it cannot be a topological surface state. Hlawenka et al. have considered this possibility and have concluded that this state is indeed an energy-shifted, surface-localized d-f hybrid that it is not confined to the bulk hybridization gap. Ohtsubo et al. do not consider this possibility at all and this is very crucial because such a scenario is incompatible

with assigning a topological character to these states. I believe that it would be fairer to consider both scenarios (in-gap topological vs. surface-confined but not in-gap) on an equal footing and the reader could judge by himself. The observation of spin polarization for S1 does not change much of the above argument as spin polarization could be very well due to spin-dependent reflectivity as for S2 and F. Therefore, as it stands now, I cannot recommend the manuscript for publication.

Now we got what the reviewer is concerning. To avoid the confusion, let us summarize what is known in the gap formation behaviour of SmB_6 mainly based on transport experiments (e.g. Flachbart PRB2001, we added it as ref. 37 in the main text).

1. Above ~ 70 K, SmB_6 is a good metal with high electrical conductivity.
2. In 70-15 K, the conductivity of SmB_6 decreases with an order of magnitude, opening an activation gap of 10-20 meV.
3. Below 10 K, the other activation gap (3-5 meV) opens.

As far as we understood, the concern of the reviewer is that the bulk electronic structure of SmB_6 at 15-20 K, the region 2 above, could not be an insulator, attributing the BOTH activation gaps (at ~ 70 K with ~ 20 meV and at ~ 10 K with 3-5 meV) to be the bulk bands. If it is correct, the topological classification performed in this work becomes nonsense, because SmB_6 is not an insulator yet at 15-20 K.

We do not think this case is likely. Firstly, it is well known that the electrical conductivity decreases with an order of magnitudes in the region 2, clearly indicating that the majority of the electronic structure in SmB_6 transforms to be an insulator with the activation gap of ~ 20 meV. In this temperature range, the thermally excited bulk carriers are still contributing to the electrical transport. However, it does not mean the bulk band gap is absent. Since the bulk metallic band (Sm 5d) at high temperature forms large Fermi surfaces, the conductivity of SmB_6 in the region 2 must be in the same order of magnitude as the high-temperature bulk-metallic phase, if the density of states from the bulk Sm 5d bands still survive at the Fermi level. This assumption is denied by the clear increase of the electrical conductivity below 70 K.

Secondly, if any bulk metallic bands, not the thermally excited carriers across the 20 meV activation gap, survive at the Fermi level, it must be observed by ARPES. Actually, it is what this reviewer has claimed with very strong tone during this peer-review process; bulk bands derived from the Sm-5d and 4f orbitals must be observed by ARPES. We agree with the reviewer on this point and thus think the claim by Hlawenka et al. would not be the actual picture of SmB_6 . In fact, not only our data but the other numerous ARPES studies with various experimental conditions (refs. 9-14) have reported no bulk bands at the Fermi level.

However, it is right that Hlawenka et al., have actually claimed so, even though they did not provide any experimental evidence for that; they found only the 2D bands at the Fermi level. For fairness, we added this point in p. 9. In short, the added paragraph tells that there is a claim, but no experimental data supporting it has been reported so far.

Reviewer #3 (Remarks to the Author):

We'd like to thank the reviewer again for the fruitful discussion to improve our MS. We are happy that he/she keeps encouraging the improvement and the subsequent publication of our MS, while pointing out some concerns. Here, we'd like to provide our answers to the reviewer's comments.

This second revision of the manuscript still has some issues to comment on.

(1)

The responses to Referee #1 and #2 concerns over the S2 and F labeling and discussion were illuminating to learn what details were missing and not discussed in earlier drafts.

Thanks to the discussion with all the reviewers, we could improve the discussion on the assignments of S2 and F. We are sure that the current presentation of S2 and F is the fairest way to introduce what were observed in this work. Please find the latest discussion also in the points (1), (2) for the reviewer #1 and (1) for the reviewer #2.

(2)

Similarly, it is very disappointing that the authors failed to properly describe their experimental setup and did not reference their current upgraded configuration of two VLEED detectors [Ref. 43] until two rounds into the review process. If they had done so earlier, then a natural question to have been asked is why do the authors not offer a complete presentation of all three spin-components for any particular k-point? Since the VLEED/exchange scattering is a serial measurement of each individual spin-component (with a 4-step serial measurement of opposite target magnetizations as now described), did the authors specifically only measured a subset of the 3 vector components or did the authors measure all three and are only presenting the subset relevant to their spin texture model? I am willing to now accept the earlier "mislabeling" answer to the previous initial draft issue of the spin-asymmetry direction in Fig. S1 spectrum. It would be nice if the authors also added the 2015 double VLEED reference [43] to their supplemental description of the experimental geometry, and in the description of Fig. S6. provided the information (if correct) that in addition to rotating the azimuth angle of the sample relative to Fig. 4(a,c,d), the "other" VLEED detector was used to obtain the "vertical" in-plane component, i.e. orthogonal to the "horizontal" component measured in Fig. 4(a,c,d). Fig. 4(b) appears to be measured with the sample azimuth rotated relative to Fig. 4(a,c,d) panels and with the same VLEED detector for the "horizontal" in-plane component.

We're afraid that we did not observe the spin polarization of the SmB_6 electronic states parallel to the in-plane wavevector. Because the SARPES measurement is quite a time-consuming one, even by using the high-efficiency VLEED spin detectors, and the machine time for the measurement is limited, we had to choose the information to be collected.

To make the topological classification, the first task for SARPES is to check whether the metallic surface state is doubly spin degenerated or not. This is the crucial information to count the number of Fermi contours correctly. For this purpose, one just need to check if the state is spin polarized or not. The orientation is not important in this stage. Therefore, we choose the most likely orientation, in-plane, helical ones for this purpose.

As a next step, we tried to detect the spin polarization characteristic of the new (111) surface and found the out-of-plane polarization reflecting the lack of two-fold rotation symmetry of the (111) surface (as explained in Fig. S7).

Then, the time was up. In this manuscript, we have provided almost all the SARPES data we had measured.

To obtain the total, three-dimensional spin polarization information and to discuss the possible interaction among them (as discussed in refs. 17, 18 in SM) with strong electron correlation would be an interesting work for the future. However, it is out of the focal point of this MS.

As for the description of the SARPES measurements, especially which VLEED detector to be used, we have added the information in p. 2 in SM according to the advice from the reviewer.

(3)

The authors did not fully understand my technical complaint concerning the summary sentence about “a consistent interpretation . . . has been achieved.” There is a general guideline in many journals (although I do not see it explicitly mentioned for this journal) that the main manuscript should stand on its own and be understandable without the supplement. There is no problem with stating that some other interpretation of the (001) surface ARPES is described in the supplement with no further details given in the main manuscript, e.g. merely “...reconciled by an interpretation...”. However if one is going to then use the results of that supplementary discussion in the conclusion summary paragraph as a key point with the very assertive “consistent interpretation . . . has been achieved”, then this sentence violates the stand-on-its-own guideline. I.e. the reader must go to the supplement to understand what this “achievement” is all about, since the sentences a few paragraphs earlier calling attention to the topic provides no specific details beyond “an interpretation”. What I am asking is that the authors be consistent with all their other references to the supplement (“SM”) in which it is stated clearly what is to be found there. I offered an example recommendation of such specific details in the previous review. Also in the similar vein of complaints about “being too assertive”, the word choice “has been achieved” should be changed “has been proposed”.

We thank the reviewer for the clear and instructive explanation. We accept it and remove the sentence "Based on these results... (001) surfaces has been achieved." from Summary.

The other assertive word choices, such as "smoking-gun evidence", are also modified mainly in the abstract and summary paragraphs. Please find the red-coloured phrases there.

(4)

The authors newly use the term “many-body resonance” as a shortcut to describe origin of the (001) in-gap states in the Hlawenka model [Ref. 14]. Those authors actually used the term “many-body resonance...shift”, i.e. the “many body resonance” refers to the 4f peak and the “shift” refers to the energetic response (in the first sub-surface layer) to the various termination-dependent surface charges. Hence it falls into the category of “surface polarity”, but distinctly different from Zhu [Ref. 36] whose model is more a combination of “dangling-band + surface polarity”. The authors should use a different descriptive term for the Hlawenka model. It is noteworthy that the relaxed (110) and (111) surfaces are not necessarily immune to similar near-surface polarity-induced energy shifts.

According to the advice from the reviewer, we added description to explain what the phrase "many-body resonance" means so that the readers could understand the model assumed there. According to the requests from the reviewers, we added a section in SM to discuss the origin of the observed bands (S1, S2 and F). We moved the discussion about the characteristics of the observed bands there (p. 13-14 in SM).

(5)

The authors now explicitly state "that the detailed origin of the surface states, such as ..., plays no role for the (topological) classification". While I understand where this statement is coming from, my opinion is that in the real world of scientific argumentation, it does matter. If in the future some new compelling evidence is presented that the (111) surface in-gaps states have a surface polarity origin, AND that origin does not have a clear theoretical basis for generating single-spin states, or has no connection to parity inversion of bulk bands, then one has a conflict that needs to be resolved. One either has to refine the origin model or maybe look again at the link between single-spin states and the origin of the measured partial spin-polarization, such as the possibility described in the newly added reference [30] of different spin-dependent reflectivities of Bloch waves off the surface that generates spin-polarization in otherwise spin-degenerate bands. In that sense, the type of surface state origin does matter.

We agree with the reviewer that the future new compelling evidence could renew the interpretation, especially the topological classification, in this work, while such work is just a future work at present, to the best of our knowledge. In the current version of MS, we have added a discussion paragraph (p. 9) in order to state the limitation and future possible update of our interpretation explicitly, as follows.

The topological classification procedure assumes weakly correlated insulator. Therefore, we cannot exclude a possible violation of such simple topological classification by strong electron correlation. To the best of our knowledge, such work has never been published so far. However, once such discovery has been achieved, the topological classification performed in this work should be revisited.

I can continue to recommend this manuscript for publication for its contribution to the on-going debate on SmB_6 , even if my confidence in the result tends to weaken after each review stage.

REVIEWERS' COMMENTS:

Reviewer #2 (Remarks to the Author):

I have read the new version of the manuscript. I appreciate the fact that the authors have started considering other interpretations and the shortcomings of their -previously unquestionable- interpretation on the origin of the metallic states by adding a new section in the suppl. info and another paragraph before the summary.

The authors understood very well my concern that the SmB₆ surface as seen by ARPES may not be a bulk insulator. They answered this concern by referring to the conclusions of transport studies. I do not question any of these results. I am simply saying that ARPES as a surface sensitive technique may see a band-bent version of the transport results. Bi-based TIs are prime examples where -for ARPES- these materials (Bi-based TIs) are not bulk insulators and they even become less and less insulating over the course of an experiment. Due to band bending we could have bulk bands crossing EF: the bands that the authors see (S1) that look like the bulk bands because they simply are the confined bulk bands after band bending. As transport is less sensitive to surface band bending this effect is not seen there. The question whether these states (S1) are really the bulk bands (as seen by ARPES) or an in-gap state still remains for me.

Having said that, I agree with Referee 1 that these results are worth publishing. I had said it since the first round. We may disagree on the interpretation but the authors have taken some steps back from their initial exaggerated confidence. I will not oppose publication in nature communications.

We would like to thank the reviewer 2 again for his/her efforts to improve our manuscript. We are happy that the reviewer 2 finally agrees with the publication of this work. Here, we answer the last concern posed by the reviewer 2. This comment and reply would be published together with the accepted MS itself. We hope this communication would also be helpful for the readers and the following researches in the issue of SmB_6 .

Reviewer #2 (Remarks to the Author):

I have read the new version of the manuscript. I appreciate the fact that the authors have started considering other interpretations and the shortcomings of their -previously unquestionable- interpretation on the origin of the metallic states by adding a new section in the suppl. info and another paragraph before the summary. The authors understood very well my concern that the SmB_6 surface as seen by ARPES may not be a bulk insulator. They answered this concern by referring to the conclusions of transport studies. I do not question any of these results. I am simply saying that ARPES as a surface sensitive technique may see a band-bent version of the transport results. Bi-based TIs are prime examples where -for ARPES- these materials (Bi-based TIs) are not bulk insulators and they even become less and less insulating over the course of an experiment. Due to band bending we could have bulk bands crossing EF: the bands that the authors see (S1) that look like the bulk bands because they simply are the confined bulk bands after band bending. As transport is less sensitive to surface band bending this effect is not seen there. The question whether these states (S1) are really the bulk bands (as seen by ARPES) or an in-gap state still remains for me.

We do not think this concern is relevant for the topological classification performed in this work. For the topological classification, the "confined bulk bands after band bending" is nothing but a surface state, because it is confined in surface layers (clear difference from the real bulk bands) and thus has 2D character without any dispersion along the surface normal. If such confined 2D state forms odd numbers of Fermi contours with spin polarization, it is the smoking-gun evidence of topologically non-trivial bulk electronic structure of the substrate. Actually, this point has been already discussed in detail in the second rebuttal letter (third round). We show it again here.

The possible origin of the surface state from projection and surface confinement is NOT the confliction against the topological order. The microscopic origin of the surface states is irrelevant to the topological classification. In other words, if the odd number of spin-polarized FCs, derived from surface confinement for example, appeared around surface TRIMs, the non-trivial topological order of the bulk electronic structure is proved by them without any conflict.

For example, the partial charges of the interface state of HgTe has a long tail to the deeper layers (e.g. Fig. 3 of PRL105, 176805). Such character is quite similar to the confined 2DEG at the LAO/STO interface (Fig. 2 of PRB 79, 245411). Both of them also shows quite similar dispersions and orbital characters to their "mother" bulk states. However, at the same time, the 2D state of HgTe is the typical topological surface state reflecting the non-trivial topological order of HgTe.

Having said that, I agree with Referee 1 that these results are worth publishing. I had said it since the first round. We may disagree on the interpretation but the authors have taken some steps back from their initial exaggerated confidence. I will not oppose publication in nature communications.